



# The urban dispersion model EPISODE. Part 1: A Eulerian and sub-grid-scale air quality model and its application in Nordic winter conditions

Paul D. Hamer[1], Sam-Erik Walker[1], Gabriela Sousa-Santos[1], Matthias Vogt[1], Dam Vo-Thanh[1], Susana Lopez-Aparicio[1], Martin O.P. Ramacher[2], Matthias Karl[2]

[1]Norwegian Institute for Air Research (NILU), Kjeller, Norway
[2]Chemistry Transport Modelling Department, Institute of Coastal Research, Helmholtz-Zentrum Geesthacht, 21502, Geesthacht, Germany

*Correspondence to*: paul.hamer@nilu.no

**Abstract.** This paper describes the Eulerian urban dispersion model EPISODE. EPISODE was developed to address a need for an urban air quality model in support of policy, planning, and air quality management in the Nordic, and, specifically, Norwegian setting. It can be used for the calculation of a variety of airborne pollutant concentrations, but we focus here on the implementation and application of the model for $NO_2$ pollution. EPISODE consists of a Eulerian 3D grid model with embedded sub-grid dispersion models (e.g., a Gaussian plume model) for dispersion of pollution from line (i.e., roads) and point sources (e.g., chimney stacks). It considers the atmospheric processes advection, diffusion, and a $NO_2$ photochemistry represented using the photostationary steady state approximation for $NO_2$. EPISODE calculates hourly air concentrations representative of the grids and at receptor points. The latter allow EPISODE to estimate concentrations representative of the levels experienced by the population and to estimate their exposure. This methodological framework makes it suitable for simulating $NO_2$ concentrations at fine scale resolution (< 100 m) in Nordic environments. The model can be run in an offline nested mode using output concentrations from a global or regional chemical transport model and forced by meteorology from an external numerical weather prediction model; but it also can be driven by meteorological observations. We give a full description of the overall model function as well as its individual components. We then present a case study for six Norwegian cities whereby we simulate $NO_2$ pollution for the entire year of 2015. The model is evaluated against in-situ observations for the entire year and for specific episodes of enhanced pollution during winter. We evaluate the model performance using the FAIRMODE DELTA Tool that utilizes traditional statistical metrics, e.g., RMSE, Pearson correlation, R, and bias along with some specialised tests for air quality model evaluation. We find that EPISODE attains the DELTA Tool model quality objective in all of the stations we evaluate against. Further, the other statistical evaluations show adequate model performance, but that the model scores greatly improved correlations during winter and autumn compared to the summer. We attribute this to the use of the photostationary steady state scheme for $NO_2$, which should perform best in the absence of local ozone photochemical production. Oslo does not comply with the $NO_2$ annual limit set in the 2008/50/EC directive (AQD). $NO_2$ pollution episodes with the highest $NO_2$ concentrations, which lead to the occurrence of exceedances of the AQD hourly limit for $NO_2$ occur primarily in the winter and autumn in Oslo, so this strongly supports the use of EPISODE in the application of these winter-





time events. Overall, we conclude that the model is suitable for assessment of annual mean NO$_2$ concentrations and also for the study of hourly NO$_2$ concentrations in the Nordic winter and autumn environment. Further, we conclude that it is suitable for a range of policy applications that include: pollution episode analysis, evaluation of seasonal statistics, policy and planning support, and air quality management. Lastly, we identify a series of model developments specifically designed to address the

limitations of the current model assumptions. Part 2 of this two-part paper discusses the "CityChem" extension to EPISODE, which includes a number of implementations such as a more comprehensive photochemical scheme suitable for describing more chemical species and a more diverse range of photochemical environments, and a more advanced treatment of the sub-grid dispersion.

## 1 Introduction

Air pollution represents a major hazard to human health. An estimated 3 million people die each year worldwide due to ambient air pollution (World Health Organization, 2016), which includes combined effects from O$_3$, NO$_2$, SO$_2$, and particulate matter (PM). Of these listed pollutants, PM has the largest impact on mortality and disease burden worldwide. 90% of the world's population breathes air that does not comply with WHO guidelines (World Health Organization, 2016). Further, human exposure to poor air quality is disproportionately weighted to populations living in urban areas where population densities,

relatively high levels of pollutant emissions and consequent high background levels of pollutants coincide spatially.

The European Commission Directive 2008/50/EC (EU, 2008) requires that air quality be monitored and assessed via measurement and/or modelling for 13 key pollutants in European cities with populations larger than 250,000 people. Measurements are required in all cases except for when pollutant concentrations are very low. In addition, Directive 2008/50/EC indicates that, where possible, modelling should be applied to allow the wider spatial interpretation of in-situ

measurement data. Norway as a European Economic Area (EEA) member adopted these regulations within its own laws.

The health impacts of urban air pollution and the requirements from legislation to provide air quality assessment and management for urban areas combine to create a need to develop urban air quality models. Such models need to provide air quality exposure mapping and to further support policy-making through assessment of emission abatement measures and understanding of the sources, causes and processes that define the air quality.

Due to the historical need and priority to assess of trans-boundary pollution (e.g., Fagerli et al., 2017), finite computational power that limits model resolution, and the resolution of the most commonly used compiled emission inventories, the majority of existing air quality models operate at a regional-scale. See, for example, the regional production of the Copernicus Atmospheric Monitoring System (Marécal et al., 2015) that includes 7 chemical transport models (CTMs) run operationally over a European domain at ~10 km resolution. In another case the CALIOPE system is being run operationally over Spain at

~4 km resolution (Baldasano et al., 2011; Pay et al., 2010) using the Community Multiscale Air Quality Modelling (CMAQ) system, and CMAQ is also being run operationally for the United States at 12 km resolution (Foley et al., 2010). The resolution of regional models means they can provide information at the background scale for urban areas, but this limits them from





providing the necessary information for policymakers (e.g., exposure mapping and assessment of abatement measures) at the urban and street scales. This limitation stems from a lack of dispersion at the scale of tens to hundreds of meters that prevents them from simulating the typically higher concentrations found close to pollution sources, which are frequently found in areas of higher population density. In addition, the gridded nature of most emission inventories specifically prevents them from

representing the actual geometry of emission sources at the sub-kilometre scale, i.e., line (along roads) and point (e.g., industrial stack emissions) sources. The widely used operational regional air quality models operating on the scale of 4-20 km resolution are therefore unsuitable for studying air quality at the urban and street scales.

Given the limitations of regional-scale air quality models outlined above, we were motivated to develop the EPISODE urban scale air quality model with the specific aim of addressing many of their weaknesses. EPISODE is a 3D Eulerian grid model

that operates as a CTM whereby meteorological fields are provided by a separate numerical weather prediction (NWP) model or by diagnostic processing of meteorological observations. EPISODE is typically run at 1 km$^2$ resolution over an entire city with domains of up to ~1000 km$^2$ in size. The Eulerian grid component of EPISODE simulates advection, vertical/horizontal diffusion, background transport across the model domain boundaries, and photochemistry. EPISODE also includes several sub-grid scale processes including emissions (line source and point sources), Gaussian dispersion, photochemistry, and,

importantly, estimation of concentrations at the sub-grid scale in locations specified by the user.

In addition to the model EPISODE, and of key importance for its practical use, historical developments in EPISODE have occurred in parallel with a GIS-based system called AirQUIS (Endregard, 2002; Sivertsen and Bøhler, 2000; Slørdal et al., 2008b, 2008a) designed to estimate emissions for line and point sources. This element of the system is essential because the inclusion of sub-grid scale emission and dispersion processes achieves nothing without, for example, having the correct line

source configuration for the real road network. AirQUIS is currently only available under commercial license.

EPISODE's typical model resolution, scale of representation (i.e., down to tens of meters), size of domain (i.e., city scale), and the level of detail of its sub-grid scale transport processes (i.e., Gaussian dispersion), place it in the gap between regional-scale air quality models on the one hand, and micro-scale models on the other. Micro-scale models represent the dispersion of pollutants to a higher level of detail compared to Gaussian dispersion by using large eddy simulation (LES) techniques, but

typically only represent limited areas in a city that therefore limit their application for wider scale studies of the urban environment.

In fact, other modelling systems have been developed for urban scale air quality modelling that have been motivated by similar needs for air quality mapping and decision support systems operating at the urban scale. These include the Danish AirGIS system (Jensen et al., 2001) that uses the street canyon air quality model OSPM, the CALIOPE-Urban system that couples the

CALIOPE regional air quality model with the urban roadway dispersion model R-LINE (Baldasano et al., 2011; Benavides et al., 2019; Pay et al., 2010), the Swedish Enviman system (Tarodo, 2003), and the Austrian Airware system (Fedra and Haurie, 1999). Each of these models follows different approaches, but suffice to say they perform a necessary role in support of air quality management and fill a niche between regional-scale air quality models and microscale models.



This article consists of two parts. Part one describes the components of the EPISODE model including Eulerian grid processes, photochemistry based on the photo-stationary state (PSS) approximation for NO, $NO_2$ and $O_3$ photochemistry, sub-grid processes, and various pre-processing utilities. Importantly, the limitations of the PSS approximation for the NO, $NO_2$, and $O_3$ chemical system limit EPISODE's application to conditions where net photochemical production of $O_3$ makes little

contribution to background $O_3$ levels. Part one, therefore, examines an application of EPISODE in the Nordic winter setting. part two (Karl et al., 2019) of this article describes the EPISODE-CityChem model, which builds upon the EPISODE model described in part one, but describes the implementation of a more comprehensive photochemical scheme that can have wider applicability including lower latitude locations. Part two describes an application of EPISODE-CityChem for the city of Hamburg.

Section 2 of this paper describes the EPISODE model and all of its components including external pre-processing utilities. Section 3 describes the case study and EPISODE model setup for seven cities in Norway. Section 4 describes the results from the case study and provides an evaluation of the model performance. Section 5 contains a summary, and Sect. 6 the future work we have planned to further develop EPISODE independent from the planned work to develop EPISODE CityChem described in part two (Karl et al., 2019).

## 2 Description of EPISODE

### 2.1 Overview of EPISODE Model Components

The EPISODE CTM simulates the emission, photochemistry and transport of $NO_x$ in urban areas with the specific aim of simulating the pollutant $NO_2$. EPISODE can also simulate the emission and transport of both PM2.5 and PM10. Currently, both PM2.5 and PM10 are treated as inert tracers with no secondary aerosol formation, but this is planned to be added in future

versions of the model (see part two for further explanation, Karl et al., 2019). EPISODE consists of a 3D Eulerian grid CTM model that has sub-grid Gaussian dispersion models for the dispersion of pollutants from both line and point emission sources embedded within it. EPISODE is capable of sampling the concentrations dispersed at the sub-grid scale using receptor points. As a result, EPISODE can provide output concentrations in the 3D grid and also at the receptor points. The user defines the location of the receptor points and practically EPISODE can be run with up to 35,000 receptor points distributed over a city.

This is not a hard limit, but significant degradation in computational performance occurs with higher numbers of points. The user has complete freedom to either define a regular grid at a fine scale, to align the receptor points near pollution sources, e.g., along road routes, or to enact some combination of both strategies.

EPISODE is driven by meteorological inputs in the Eulerian 3D grid. The meteorological inputs can either be provided by a separate NWP (e.g., AROME or WRF – both used in past and current projects), or from one of the two meteorological pre-

processing utilities developed for use with EPISODE described in Sect. 2.3. These meteorological inputs drive the transport processes at both the grid and sub-grid scales.





Emissions in EPISODE can be emitted either as area gridded emissions and/or as sub-grid emissions, i.e., line and point sources. The emissions inputs can be setup in a fully customisable manner such that emissions from a single sector or sub-sector can be emitted as either area gridded or sub-grid emissions, or in fact as both types if so desired. In practice, the choice to emit a pollutant as area gridded or sub-grid emissions depends on the specific application of the EPISODE model, and also

the level of detail that exists on the spatial distribution for any particular emission sector.

The Eulerian 3D grid model (described in Sect. 2.2.1) consists of an advection scheme, vertical and horizontal diffusion schemes, and area gridded emissions. We also discuss the topography inputs and the surface roughness inputs. The Eulerian grid model also includes the treatment of the initial and boundary conditions from background concentrations of pollutants, and the photo-stationary state scheme for $NO_2$, NO, and O3 chemistry.

The sub-grid model in EPISODE (Sect. 2.2.2) consists of line and point source sub-grid emissions, and following from that both line and point source dispersion. The last component of the sub-grid model consists of sampling of the Gaussian dispersion at user specified receptor points. Note that the solution to the PSS for $NO_2$, NO, and $O_3$ is also calculated at each of the receptor points. This is done in such a way that it combines the calculated NO and $NO_2$ dispersed from the point and line sources with the background $O_3$ calculated at the relevant grid for each receptor point.

Figure 1 provides an overview of each of the model components, i.e., model inputs, processes, etc., and how they interact with one another.

## 2.2 Description of Individual Model Components

### 2.2.1 Eulerian Grid Model

The EPISODE model simulates the emission, 3D advection, diffusion, chemical reaction (in the case of $NO_2$), and the transport

into and out of the model domain boundaries in the urban atmosphere on a 3D Eulerian grid. The model horizontal gridding is specified in Universal Transverse Mercator (UTM) coordinates typically at 1 $km^2$ resolution. The vertical grid is a terrain-following sigma coordinate system defined from an idealised hydrostatic pressure-distribution. EPISODE is typically run with a relatively high vertical resolution for a CTM with a surface layer thickness of only between 19 and 24 meters in height, and the layers immediately above this first layer are often of comparable thickness. This helps EPISODE to represent higher

concentrations in the surface layer and to represent finer scale variability in concentrations in the vertical. The extent of the grid in the vertical used in current applications is typically up to 4000 meters at most, but this is not a hard limit.

The resolution of the spatial gridding in EPISODE has constraints applied on it arising from the equations governing the transport on the Eulerian grid. The terms describing the vertical turbulent diffusion are represented according to the mixing length theory (K-theory). K-theory is only applicable as long as the chemical reaction processes are slow compared to the

speed of the turbulent transport. This condition is not satisfied only in cases with extremely fast chemical systems, e.g., oxidation of monoterpenes above forest canopies. The $O_3$ and $NO_x$ chemical system is sufficiently slow for this condition to be satisfied. In addition, the characteristic length scales and time scales for changes in the mean concentration field must be





large compared with the corresponding scales for turbulent transport (Seinfeld and Pandis, 2006), e.g., at the scale at which large eddies are resolved. The validity of K-theory at small spatial scales, therefore, places a limit on the spatial resolution of the Eulerian main grid in EPISODE. In our applications here, we use a horizontal resolution of 1 km, which should be well above the limitation created by these issues.

The pollutant concentrations are calculated by integrating forward in time the solutions for the 3D advection, diffusion, and photochemistry equations. The transport of pollutants in and out of the model domain is implicitly considered within the 3D advection equations. The numerical solution for the Eulerian grid employs operator splitting to separately solve the processes: (1) advection; (2) diffusion; and (3) chemical reactions. The derivation of the sigma-coordinate transform of the advection/-diffusion equation is described in the technical report Slørdal et al. (2003).

EPISODE's numerical model time step is calculated dynamically based on the critical time steps associated with the solution of the 3D advection and diffusion processes. This is done in such a way that the shortest critical time step across the three processes is selected and then applied for each process, including the PSS chemistry for $NO_2$, NO, and $O_3$ at the grid-scale. The time step is also adjusted slightly downward to ensure that nsteps = 3600(s)/dt is always an integer value. This way, all operations are performed an even number of times so that every other operator sequence may be a mirror in the opposite

direction of the first sequence to reduce time-splitting errors. The dynamical timestep typically has a duration of a few minutes. Different schemes have been developed for both the 3D advection and diffusion transport processes (see Table 1), and for other processes on the 3D grid such as for the treatment of background pollutant concentrations (see Table 2). These different schemes are described in the sections that follow.

**3D Advection Schemes**

Advection terms are used in EPISODE to represent both bulk transport both in the horizontal and the vertical. In the vertical dimension the advection term encompasses bulk vertical transport arising from convection and other bulk vertical motion that are represented at the grid-scale in the input wind fields. Two different horizontal advection schemes have been implemented in EPISODE along with a single scheme for vertical advection scheme. The first advection scheme is an implementation of (Bott, 1989, 1992, 1993) consisting of a 4th-order positive definite scheme. The scheme calculates fluxes between the grid cells

based on a local area-preserving 4th-degree polynomial describing the concentration fluctuations locally. The Bott scheme (1989, 1992, 1993) has good numerical properties and very low artificial numerical diffusion. It employs a time splitting method whereby advection is solved separately in the x and y directions with the order of operations for the x and y-axes alternating every second timestep. This scheme is used in every current application of the EPISODE model.

The second advection scheme is a variation of the first Bott scheme and consists of a 4th-order positive definite and monotone

scheme. This implementation of the Bott scheme has only been used experimentally in EPISODE.

EPISODE has various methods for specifying the boundary conditions for background concentrations (see the relevant section further in Sect. 2.2.1). For each of these methods and except for the first time step (in which case background concentrations are set as the initial concentrations in all the model domain), the background concentrations are specified in grid cells





enveloping the model domain (with the same horizontal and vertical resolution) bordering it in the x, y, and z dimensions. Background concentrations are specified in each of these grid cells at every time step. The background concentrations in these grid cells are included in the solution for the advection, and it is by this mechanism that background concentrations are transported into the domain. Imposing a background concentration in the boundary grid cells can result in a problem of spurious

wave reflections at the inflow/outflow boundary. This problem is addressed via a modification of the Bott's scheme for advection near the boundaries. A 1st order polynomial is used in the model domain grid cells bordering the boundary with the grid cells with background concentrations, i.e., [1, y], [X,y], [x, 1], or [x,Y] (X and Y represent the last grid cells in the x and y dimension), to compute the fluxes in and out of the model domain across the boundary. A $2^{nd}$ order polynomial is used in the second cells of the model domain from the boundary, i.e., [2, y], [X-1, y], [x, 2], or [x, Y-1]. The Bott scheme $4^{th}$-order

polynomial is used in the third cells of the model domain from the boundary, i.e., [3, y], [X-2, y], [x, 3], or [x, Y-2] and the other cells of the inner model domain. As a test of the model's treatment of boundary conditions, the entrainment of ozone and $PM_{2.5}$ from the boundaries into the inner domain for constant wind direction was studied in an artificial simulation in Appendix D of part two of this article (Karl et al., 2019).

Vertical advection is calculated using the simple upstream method, which has the property of being strongly diffusive.

However, this numerical diffusion is insignificant in comparison to the magnitude of the vertical turbulent diffusion term.

The topography within the domain is defined on the Eulerian grid in terms of the average elevation above sea level in meters. The topography information, therefore, shares the same resolution as the model. It is specified as an input file to the model in ASCII format and can either be defined according to mapping information or can be specified as a constant across the domain.

**Vertical and Horizontal Diffusion Schemes**

The values of the eddy diffusivities depends on the spatial structure of the flow field, which is difficult to solve in the grid resolution used here. Therefore, both the horizontal and vertical eddy diffusivities need to be calculated on the EPISODE Eulerian grid with parameterisations using the bulk 3D grid meteorological variables as input variables. The transport of pollutants in the vertical direction is often dominated by turbulent diffusion. The parameterisation of the vertical eddy diffusivity, therefore, has important consequences for the vertical profiles of pollutant concentrations.

In the case of horizontal diffusion, a single parameterisation scheme has been implemented that consists of the fully explicit forward Euler scheme (Smith, 1985).

In EPISODE, the model user can choose between two different parameterisations of the vertical variations of vertical eddy diffusivity, $K^{(z)}$: (1) the standard $K^{(z)}$ method, which is the default in the EPISODE dispersion model; and (2) the new urban K(z) method, which has been newly implemented in the EPISODE model. These are both described below. Both

parameterizations depend on the atmospheric stability of the Planetary Boundary Layer (PBL). The stability regime (related to atmospheric buoyancy in the PBL) affecting these $K^{(z)}$ methods is defined with a non-dimensional number z/L, where z is the height above the ground and L is the Monin-Obukhov length. In accordance with K-theory, it is assumed that chemical species have non-dimensional profile characteristics similar to potential temperature, θ, such that $K^{(z)}$ equals the eddy





diffusivity of the heat flux. In order to model the turbulent processes in the PBL in a realistic manner, it is essential to consider the vertical variation of the exchange coefficients. Therefore, in the explicit closure schemes used here, a profile is prescribed to account for the vertical variation of the turbulent exchange coefficients.

The applied vertical eddy diffusivity, $K^{(z)}$, is defined as a sum of two terms:

$$K^{(z)} = K_*^{(z)} + K_0^{(z)}, \qquad (1)$$

where $K_*^{(z)}$ is a parameterisation depending on stability regime and $K_0^{(z)}$ is an added background diffusivity term. $K_0^{(z)}$ is only applied within the boundary layer.

The standard $K^{(z)}$-method is based upon the description given in Byun et al. (1999) and included in Sect. S1 of the Supplement. The standard $K^{(z)}$-method uses a constant background diffusivity of $K_0^{(z)} = 0.01$ m$^2$ s$^{-1}$.

We now describe the new urban $K^{(z)}$ method here in the main text. For neutral conditions the expression from Shir (1973) is adopted:

$$K^{(z)} = \kappa u_* z \exp\left(\frac{8fz}{u_*}\right), \qquad (2)$$

where $\kappa = 0.41$ is the Von Kármán constant, $u_*$ is the friction velocity (ms$^{-1}$) and $f$ is the Coriolis parameter.

For unstable conditions, we use the complex polynomial expression by Lamb and Durran (1978), which is applied as a component within a more comprehensive scheme in McRae et al. (1982).

For stable conditions, a modified equation by Businger and Arya (1974) is used. Businger and Arya (1974) developed a steady state, first-order numerical K$^{(z)}$-model based on a non-dimensional eddy viscosity derived from the empirical log-linear profile for the stable atmospheric surface layer. In this equation, the temperature gradient parameterisation from Businger et al. (1971) is replaced by the non-dimensional temperature gradient ($\Phi_H$) given by Beljaars and Holtslag (1991):

$$\Phi_H = 1 + \frac{z}{L}\left[\alpha\sqrt{1 + \frac{2}{3}\frac{\alpha z}{L}} + \beta e^{-\delta\frac{z}{L}}\left(1 + \gamma - \delta\frac{z}{L}\right)\right], \qquad (3)$$

where the suggested values of the empirical coefficients are: $\alpha = 1$, $\beta = 2/3$, $\gamma = 5$, and $\delta = 0.35$. The expression of Businger and Arya (1974) for the vertical eddy diffusivity under stable conditions consequently becomes:

$$K_*^{(z)} = \frac{\kappa u_* z}{0.8\left(1 + \frac{z}{L}\left[\alpha\sqrt{1 + \frac{2\alpha z}{3L}} + \beta e^{-\delta\frac{z}{L}}\left(1 + \gamma - \delta\frac{z}{L}\right)\right]\right)} exp\left(\frac{8fz}{u_*}\right). \qquad (4)$$

Note that the expression from Beljaars and Holtslag (1991) is scaled by 0.8 to be in better agreement with the temperature gradient from LES computations of the stable boundary layer made by Basu and Porté-Agel (2006).

The new urban $K^{(z)}$-method, considers a baseline turbulent mixing due to the urban roughness and anthropogenic heating effect in cities, with an apparent eddy diffusivity of (Slørdal et al., 2003):

$$K_*^{(0)} = \begin{cases} (2\,\Delta z_1)^2/3600s & \text{for } u_* > 0.2 \text{ ms}^{-1} \\ (\Delta z_1)^2/3600\,s & \text{for } u_* > 0.1 \text{ ms}^{-1} \end{cases}, \qquad (5)$$

with a linear variation of $K_0^{(z)}$ between the two $u_*$ limits.



The particular choice of $K_0^{(z)}$ is based on a scale analysis. This analysis assumes that the respective minimum values of $K^{(z)}$ should be large enough, during a one hour period, to mix an air-column with a thickness of $\Delta z_1$ or $2\,\Delta z_1$, (thickness of the surface layer, i.e., the lower-most model layer), when $u_*$ is less than 0.1 ms$^{-1}$ or larger than 0.2 ms$^{-1}$, respectively (Slørdal et al., 2003). For $u_*$ less than 0.1 ms$^{-1}$ and $\Delta z1 = 20$ m, $K_0^{(z)}$ becomes equal to 0.11 m$^2$ s$^{-1}$. For $u_*$ greater than 0.2 ms$^{-1}$ and $\Delta z1$

$= 20$ m, $K_0^{(z)}$ becomes equal to 0.44 m$^2$ s$^{-1}$. A test of the two different parameterisations for the vertical eddy diffusivity is presented in Sect. 4.

The dimensionless parameter, surface roughness, z0, is required by the vertical diffusion schemes to help calculate the extent of the vertical turbulent mixing. Surface roughness has to be specified on the Eulerian grid used in the model within an ASCII input file. Surface roughness can either be specified as a constant across the whole domain, it can be specified according to an

external map of the land cover type across the domain, or the surface roughness can be imported from the NWP into EPISODE.

**Area Gridded Emissions**

Emissions in the EPISODE model can be input directly into the 3D Eulerian grid as area source emissions. In this case, emission inputs have to be specified on the EPISODE model domain grid at the working resolution of the model for every hour of the simulation. The units of the emissions are in g s$^{-1}$, and in the case of NO and NO$_X$ this is in terms of the mass of

NO$_2$ equivalent. The input format for the area source emissions is ASCII. The EPISODE model also supports full customisability for the injection heights of the area gridded emissions; the user retains the same emission input format in each file but changes the model setup so that the base emission inputs are read into the model and then emitted in the specified layers in proportions defined by the user.

EPISODE is typically run using either top-down or bottom-up emissions that undergo pre-processing to set the temporal

variability (hourly, daily, and weekly) in the emissions. This pre-processing step can be customised using the compiled statistics in Norway to inform the imposed temporal variability. However, this pre-processing step can also be customised.

**Boundary and Initial Conditions from the Pollutant Background Concentrations**

Three options exist (see Table 2) for the specification of pollutant initial and boundary conditions in EPISODE. The first option is to specify a single background concentration at all locations both in the model domain (for initial conditions) and in the grid

cells with background concentrations adjoining the model domain. In this case, concentrations can be specified to be time-varying on an hourly basis (only recommended in specific instances), or to remain constant in time (only recommended for testing purposes). This option would be perhaps used in a situation when only a single background observation station existed near a city that could be used in order to create a time series for a pollutant. The time-varying background concentration is specified in an ASCII input file while the time-invariant concentration is specified in the EPISODE run file.

The second option is to specify a vertical profile of background concentrations that is used in each grid cell in the horizontal domain and in the adjoining background grid cells. In this case, the vertical profile must have a vertical resolution identical to the EPISODE model's configuration. Similarly to the previous option, this can be done in such a way that the input profile is specified on an hourly basis or to remain constant in time. Again, the latter option is not recommended except for testing





purposes, but the time-varying option would be appropriate when the background concentrations are defined by a regional or global CTM with coarse horizontal resolution, i.e., > 50 km. The temporally varying vertical profiles can be specified in an ASCII input file whereas the constant vertical profile is specified in the EPISODE run file.

The last option gives the ability to specify background concentrations for the boundary and initial conditions on the 3D-grid

of the EPISODE model. In this case, the concentrations for the initial and boundary conditions are specified on a grid of the same horizontal and vertical resolution as the EPISODE 3D grid, and the boundary condition grid extends outside of the model domain in the x, y, and z dimensions. The background concentrations are specified on an hourly basis in NetCDF or ASCII input files. This third option in EPISODE gives the opportunity to run EPISODE in a one-way nesting configuration embedded within a regional-scale CTM. So far, this option has been used in three different configurations both using different regional-

scale CTMs to provide the fields of pollutant background concentrations. In the first example, outputs from the Copernicus Atmospheric Monitoring Services (CAMS) regional production (Marécal et al. 2015) were interpolated from their 10 km horizontal resolution down to a resolution of 1 km and also an interpolation in the vertical to achieve a vertical resolution matching that of EPISODE. This configuration has been used in the NBV and BedreByLuft projects in Norway. In the second example, output from the EMEP CTM model (Simpson et al., 2012) has also been used in similar fashion to provide

background concentrations. In the third example, the CMAQ model (Byun and Schere, 2006) was used to provide the background concentrations where the CMAQ output was at 4 km horizontal resolution and bi-linear interpolation was used to increase the horizontal resolution of the background concentrations to ~1 km. CMAQ has been used in the example presented in part two of this article (Karl et al., 2019).

**Photo-stationary State Scheme**

EPISODE has been specifically designed to be used in urban environments at high latitudes. Under these conditions that are polluted (in terms of $NO_x$) and that have relatively low levels of sunlight, it is possible to make some simplifying assumptions about the photochemistry governing the pollutant $NO_2$.

Only a small fraction of $NO_x$ emitted from motor vehicles and combustion sources is in the form of $NO_2$, the largest fraction being NO. The majority of ambient $NO_2$, therefore, originates from the subsequent chemical oxidation of NO. Under polluted,

low-light conditions, the vast majority of this oxidation occurs via reaction with $O_3$ (R1).

$$NO + O_3 \rightarrow NO_2 + O_2 \quad \text{(R1)}$$

$NO_2$ is a photo-labile molecule and readily undergoes photolysis via (R2).

$$NO_2 + \text{h}\nu \rightarrow NO + O(^3P) \quad \text{(R2)}$$

Even at the latitude of Oslo, $NO_2$ can have a lifetime with respect to photolysis on the order of minutes at midday in winter.

The reaction R2 and the subsequent reformation of O3 via (R3) must therefore be considered if we want to reasonably describe $NO_2$ concentrations under these conditions.

$$O(^3P) + O_2 \rightarrow O_3 \quad \text{(R3)}$$





Reaction (R3) between the oxygen radical (O($^3$P)) and molecular oxygen (O$_2$) occurs very rapidly and can be assumed to occur instantaneously. From this assumption, we can then reduce the photochemical system describing NO$_2$, NO, and O$_3$ under these conditions to the equilibrium reaction described in equation R4.

$$NO_2 + h\nu \leftrightarrow NO + O_3 \quad \text{(R4)}$$

Whereby the forward reaction describes the production of NO$_2$ via the reaction (R1) (with reaction coefficient k$_{(O3 + NO)}$), and the backward reaction (whose rate coefficient is assumed to be described by $J$NO$_2$) consists of the combined photo-dissociation of NO$_2$ (via (R2)) and the subsequent, assumed, instantaneous formation of O$_3$ (via (R3)). We assume that this photochemical mechanism adequately describes NO$_2$ chemistry in polluted Nordic wintertime conditions when net photochemical production of O$_3$ and losses of NO$_x$ via nitric acid production are at a minimum. However, when the solar ultraviolet (UV) radiation is

stronger, in particular during summer months or at more southerly locations, net ozone formation may take place in urban areas at a certain distance from the main emission sources (Baklanov et al., 2007). Please refer to part two of this article where the EPISODE-CityChem model is described, which uses a more comprehensive photochemical scheme suitable for more sunlit environments.

The PSS is used to resolve the NO$_2$, NO, and O$_3$ photochemistry on the 3D Eulerian grid and at the receptor points for the sub-

grid scale model. The PSS is an analytical mathematical solution that can be applied to the chemical system described in R4 to estimate the concentrations of NO$_2$, NO, and O$_3$. The PSS has two key assumptions. First, the chemical system in question is in equilibrium, and, second, that this equilibrium is attained instantaneously. These assumptions imply that the residence time of pollutants is much larger than the chemical reaction time scale. These assumptions are valid for polluted urban conditions. Section 2 in the Supplement gives an in-depth explanation of the PSS and how it is applied in this case for R4.

Taken together, the PSS and its application to R4 is therefore adequate for the Nordic case studies we present in this paper, and also for the various other previous and existing applications of the EPISODE model in Norway.

### 2.2.2 Sub-Grid Scale Model Components

**Line and Point Source Emissions**

We describe here the implementation of the sub-grid scale emissions in the EPISODE model. The line source and point source

emissions used in EPISODE are prepared ahead of running the model by one of two possible pre-processing utilities. These utilities are described in detail in Sect. 2.3. Each of these tools prepares the emission files as inputs in ASCII format that can then be read directly into the EPISODE model at run time.

The line source emission files consist of two separate files. The first contains metadata describing the road network, which is formed by road sections of variable length, width, and slope that we name road links. The second contains the hourly total

emission intensity along each road link, E$_R$, in terms of gs$^{-1}$.m for each time step of the simulation. Road link emissions are assumed to be evenly distributed along a single road link.

The point source emissions in EPISODE are used for describing emissions from stacks. The point source emission files contain the following information for each stack: their hourly emission rates in gs$^{-1}$, the geographical location of the stack in UTM





coordinates, the building width and height, the stack height and diameter, the temperature of the plume gas, and the speed at which the plume is expelled from the chimney. EPISODE reads in this information at runtime and then calculates the injection heights for the point source emission using a parameterisation based on (Briggs, 1969, 1971, 1974, 1975) that considers the processes of stack downwash, and buoyancy-driven plume rise under different stability conditions.

The stack downwash process modifies the physical height of the chimney to estimate an effective stack height (Briggs, 1974). Buoyancy-driven plume rise will affect the final plume height in different ways according to the boundary layer stability conditions, and therefore there are different parameterisations for either unstable and neutral conditions or stable conditions. The final injection height is calculated by taking into account the effects of the adjacent building (considering its height and width) on building-induced disturbances of the plume flow, plume penetration through elevated stable layers, and topography.

Further details of the parameterisations are described in S3 of the supplement.

**Line Source Gaussian Dispersion**

The line source model in EPISODE is based upon the steady-state integrated Gaussian plume model HIWAY-2 (Petersen, 1980). Line sources are defined along road links of width, $w$, and length, $D$, with a fixed rectangular area of influence surrounding each link. The area of influence is the zone within which emissions from line sources are assumed to affect

concentrations at receptor points within a single dynamical timestep. Figure 2 shows an illustration of the area of influence around an example road link. The boundaries of the distance of influence extend $R_{inf}$ (the influence distance) from the road link centres perpendicular to the road link direction. In the longitudinal direction, the distance of influence extends $R_{inf}$ from the two ends of each road link. The area of influence excludes receptor points assumed to be on the road links themselves, which is defined by the distance $R_{min}$ (Figure 2). $R_{min}$ is 5 meters plus half the road link width.

HIWAY-2 resolves the dispersion from the line sources by splitting each road link up into smaller line source segments and then calculating the dispersion from these segments individually. The line source segments are of equal length and are spaced equally along the road links. The emission intensities from each segment, $E_l$, are calculated as a fraction of the total emission along the road link, $E_R$ , according to

$$E_l = E_R \times \frac{D_l}{D_R} \qquad (6)$$

where $D_l$ is the length of the line source segment and $D_R$ is the total length of the road link. Therefore, all of the segments emit equal pollutant mass, which is proportional to the fractional length of the road segment $D_l/D_R$. Note that $E_l$ has units of $gs^{-1}$ whereas $E_R$ has units of $gs^{-1}$.m.

HIWAY-2 only calculates the dispersion from the line sources to each of the receptor points within their respective areas of influence during the last dynamical timestep of each hour. Note that the EPISODE model only outputs pollutant concentrations

on an hourly basis. Prior to the last dynamical timestep, line source emissions are only emitted directly into the Eulerian grid (see the relevant section further in Sect. 2.2.2). The implicit assumption is that due to the short transport distance, emissions from road links can only affect receptor point concentrations within the distance of influence, $R_{inf}$, on short timescales equivalent to a single dynamical timestep. The length of the dynamical timestep scales with the wind-speed such that higher





wind speeds result in shorter dynamical timesteps. The user can set the R$_{inf}$ for each road link, but typically a value of 300 m is used. That is the R$_{inf}$ used in the case study in this paper, which corresponds to a value well below the simulated distance typically travelled by an airmass in a single dynamical timestep.

The line source dispersion model is described in further detail in S4 of the supplement.

**Point Source Gaussian Dispersion**

Two point source plume parameterisations have been implemented in EPISODE to represent dispersion from chimney stacks. The first scheme is a Gaussian segmented plume model called SEGPLU (Walker and Grønskei, 1992) following the general method described by (Irwin, 1983). The second scheme is a puff model called INPUFF (Petersen and Lavdas, 1986). Both schemes use point source emissions and their injection heights calculated following Briggs (1969, 1971, 1974) described

earlier in Sect. 2.2.2 and S3 of the supplement. The emissions from point sources are treated as a sequence of instantaneous releases of a specified pollutant mass that each then, in turn, becomes a discrete puff or plume segment. The subsequent position, size and concentration of each plume segment/puff is then calculated in time by the model during each dynamical timestep. This information is then used to calculate each plume segment/puff's contribution to the receptor point surface concentrations during the last dynamical timestep of each hour.

Plume segments and puffs are no longer traced by the model during any dynamical timestep in the following cases: (1) they are transported outside of the model domain; (2) they become too large; (3) they encounter a large change in wind direction causing them to become spatially separated. If the segments or puffs become too large or are separated whilst within the model domain, then the pollutant mass within them is transferred to the grid or grids in which they currently reside during that dynamical timestep, else they are deleted (see further in Sect. 2.2.2 for more details).

The SEGPLU and INPUFF models are described in further detail in S5 and S6 of the Supplement, respectively.

**Receptor Point Concentration Calculation**

The concentrations at receptor points are calculated by combining the concentrations at the surface layer of the Eulerian grid with the contributions from line sources and point sources. The receptor point concentrations are calculated at the end of each hour. Up until that timestep, the model only calculates the chemistry and transport on the Eulerian grid, while also

simultaneously calculating the position and concentration of plume segments/puffs. The receptor point concentration at the end of each hour can be described by equation (7),

$$C_{rec}^t(r^*) = C_m^{t-1} + \sum_{l=1}^L C_{line,l}^t + \sum_{p=1}^P C_{point,p}^t \quad (7)$$

where $C_{rec}^t(r^*)$ is the receptor point concentration at receptor point $r^*$, at time $t$, $C_m^{t-1}$ is the Eulerian grid concentration from the penultimate dynamical timestep during each hour (for the grid cell x,y,z=1 where $r^*$ is located), $C_{line,l}^t$ is the line source

segment concentration contribution from line source segment $l$, and $C_{point,p}^t$ is the point source concentration contribution from a plume segment/puff, $p$. To resolve equation (7), EPISODE sums up the concentration contributions from the total number of line source segments, $L$, within R$_{inf}$ distance of the receptor point, and the total number of point sources $P$. The Eulerian grid concentration from the penultimate dynamical timestep, $C_m^{t-1}$, is used to prevent double counting because it does not include





line and point source emission contributions from the final, and current, dynamical timestep in the hour. Testing (not shown) demonstrates that using this assumption in combination with an $R_{inf}$ of 300 m (see earlier in Sect. 2.2.2) reliably reduces double counting of emissions to negligible levels.

For the simulation of the pollutant $NO_2$, the EPISODE model resolves Eq. (7) for both NO and $NO_2$, thus calculating $C_{rec}^t(r^*)$

for both compounds. Using the Eulerian grid concentration of ozone combined with the NO and $NO_2$ receptor point concentrations, EPISODE then solves the photochemistry at each receptor point using the PSS and creates an updated set of concentrations for $NO_2$, NO, and ozone at the receptor point that are provided as the hourly model outputs.

**Interaction Between Receptor and Eulerian Grid Concentrations**

Until the final dynamical timestep of the hour, the emissions from line source segments are emitted directly into the grid in

which they reside during each timestep. Each line source segment in a Eulerian grid cell (x,y,z) makes a contribution to the Eulerian grid concentration, $C_m$, which can be described as a tendency, $dC_{m,L^*}/dt$, via

$$\frac{dC_{m,L^*}}{dt} = \sum_{l^*}^{L^*} \frac{E_{l^*}}{V_{(x,y,z)}} \quad \textbf{(8)}$$

where $V(x,y,z)$ is the volume of Eulerian grid cell (x,y,z) into which the emissions occur, and $dt$ is the length of the dynamical timestep. Since we are discussing line segments within a specific grid cell here we use a specific and distinct notation different

from that in Eq. (7). Therefore, a line source segment in a particular grid cell (x,y,z) is denoted as, $l^*$, and the total number of line segments in a grid cell as $L^*$. In practice, the emissions from road links are emitted directly into the lowest layer of the Eulerian grid. Line segments are sufficiently short in length that it can be considered that each one can emit entirely within a single Eulerian grid cell.

The change in grid concentration, $\Delta C_{m,L^*}$ , due to line source segment contributions is calculated via

$$\Delta \boldsymbol{C}_{m,L^*} = \frac{d\boldsymbol{C}_{m,L^*}}{dt} \times d\boldsymbol{t} \quad \quad \textbf{(9)}$$

In the last dynamical timestep of the hour, pollutants from line sources are both emitted directly into the Eulerian grid according to (8) and are also dispersed to the receptor points according to descriptions earlier in Sect. 2.2.2 and S4 of the supplement.

Point source emissions can also contribute to both the concentrations at receptor points and the Eulerian grid. Point sources continually emit plume segments or puffs every dynamical timestep, and the plume segments/puffs are dispersed and advected

according to Sect 2.2.2.3 and S5 and S6 of the supplement. At the end of each hour, plume segments/puffs are assessed to see if their extent co-locates with receptor points at the surface, and in this case, they contribute to the receptor point concentrations via Eq. (7). Providing that the plume segments/puffs do become invalid by extending too far in the horizontal, encounter sudden changes in wind direction, or that they become separated, they will remain on the model sub-grid and only contribute to the receptor point concentrations. However, in the case that they do become invalid, they will be deleted, and the pollutant

mass within them, $m_p$, will be added to the concentration of the grid cell in which they reside as a tendency specific to that plume segment/puff, $dC_{m,p}/dt$. This tendency is calculated via

$$\frac{dC_{m,p}}{dt} = \frac{m_p}{V_{(x,y,z)} \times dt} \quad \textbf{(10)}$$





and the change in grid concentration, $\Delta C_{m,p}$, resulting from the deleted plume segment/puff mass is calculated via

$$\Delta C_{m,p} = \frac{dC_{m,p}}{dt} \times dt \quad (11)$$

## 2.3 Pre-Processing Utilities

Several pre-processing utilities are used in conjunction with the EPISODE model. These utilities are used for preparing
meteorological inputs, emissions files, and boundary condition files used in the running of an EPISODE simulation. The pre-processing utilities are:

1. MCWIND (utility to generate a diagnostic wind field);

2. CAMSBC (collection of routines to convert CAMS regional production to EPISODE background input);

3. UECT (interface for line source, point source, and area source emissions allows use of EPISODE independent of AirQUIS);

4. TAPM4CC (interface to convert TAPM meteorology output);

5. Utilities to generate auxiliary input.

Table 3 gives an overview of the purpose of the pre-processing utilities as well as outlining the input and output formats and descriptions.

The CAMSBC routines process GRIB2 and NetCDF files output from the CAMS regional production, and interpolate them
onto the chosen vertical and horizontal grid in EPISODE. In this case, the user is required to download the CAMS data directly from the CAMS online data portal (CAMS online data portal: https://atmosphere.copernicus.eu/data, last access: 16 May 2019).

The UECT routines format emissions data into the required format for EPISODE. The user must provide suitable emissions data for the city being studied. We provide some example emission inputs with the EPISODE model code to allow potential
users to understand the requirements for simulating a city.

TAPM4CC processes the 2-D and 3-D fields of meteorological variables computed by the TAPM model using an inner model grid for TAPM, which has the same model domain extent and horizontal resolution. TAPM's built-in tool TAPM2OUTA creates an *.outa file, for the time range of interest (e.g. one month), which is then used in TAPM4CC to convert the meteorological output from TAPM into binary input files for EPISODE. TAPM4CC also creates the topography input file.

The Urban Emission Conversion Tool (UECT) is a utility for preparing emission input files with hourly emission values based on the yearly emission table for the city being studied. The UECT routines convert (and interpolate) emission data into the required format for EPISODE. UECT requires emission data of geo-referenced yearly emission totals for nitrogen oxides ($NO_x$), non-methane volatile organic compounds (NMVOC), carbon monoxide (CO), sulphur dioxide ($SO_2$), ammonia ($NH_3$), as well as particulate matter $PM_{2.5}$ and $PM_{10}$, defined for each source. Missing emission totals can be indicated by "-999" in
the input to UECT. The geo-reference for point sources is the (x,y)-coordinate (in UTM) of the emission point. The geo-reference for traffic line sources is the start (x,y)-coordinate together with the end (x,y)-coordinate of the line, and the geo-reference for area sources is the (x,y)-coordinate of the lower left (southwest) corner together with the (x,y)-coordinate of the





upper right (northeast) corner of the quadratic area cell. Area sources have to be located within a regular Cartesian grid, i.e. when area sources from an ArcGIS polygon shape are used; these have to be intersected first with a raster grid.

We provide some example emission inputs with the EPISODE model code to allow potential users to understand the requirements for simulating a city.

The MCWIND utility produces a diagnostic wind field and other meteorological fields for the defined model grid, by first constructing an initial first-guess wind field based on the measurements of the horizontal wind and vertical temperature differential at two or more meteorological stations. Then the horizontal 2-D fields are interpolated to the 3-D grid of the model domain by applying Monin-Obukhov similarity theory. Finally, the first-guess 3-D wind field is adjusted to the given topography by requiring the resulting wind field in each model layer to be non-divergent and mass-consistent.

The remaining auxiliary routines are used to generate the topographic and surface roughness input from the AROME meteorological NetCDF input files and also create constant fields of both parameters of the model domain if needed.

## 3 Case Study Description and Model Setup

As a demonstration of the EPISODE model's capabilities, and as a test case for validation, we carry out the simulations of NO$_2$ concentration levels over six Norwegian cities. The chosen urban areas are Oslo, Trondheim, Stavanger, Drammen,

Grenland (including the city of Skien), and Nedre Glomma (encompassing both Fredrikstad and Sarpsborg on the Glomma river). The EPISODE model is run for the entire year of 2015 using meteorological input from the AROME model, which was run operationally over the six city domains by the Norwegian Meteorological Institute (Denby and Süld, 2015). The AROME model simulations are carried out at 1 km$^2$ horizontal spatial resolution on the exact same gridding and domain as the EPISODE model simulations for each city. The meteorological fields used in EPISODE are documented in S7 of the supplement.

AROME provides NetCDF files for input, and the surface roughness and topography used in AROME were extracted from these files.

The NO$_x$ emissions used for the simulations for each of the six city domains were developed as part of the Nasjonal Beregningsverktøy (NBV) project (Tarrasón et al., 2017). The methodologies for the creation of the emission datasets are described in (Lopez-Aparicio & Vo, 2015). The data sources, the methodology used, and emission reference years are

summarized in

Table 4 for each emission sector.

Different approaches were used to compile the emission datasets depending on the data availability for the specific emission

sector. On-road traffic emissions are estimated based on a bottom-up traffic emission model. The traffic emission model produces emissions for each road link. It takes into account the traffic volume (i.e., average daily traffic, ADT), and the heavy duty fraction of the traffic on specific road types (e.g., highway, city street, etc.). In addition, the emission model considers the





road slope. This information is obtained from the Norwegian Road Administration. The ADT is combined with temporal profiles of daily traffic to obtain hourly ADT at the road level. The vehicle fleet composition is defined as a fraction of each vehicle technology class (EURO standard) and fuel type, which combined with the HBEFA emission factors and the hourly fraction of ADT results on emissions on each road segment. The information regarding the vehicle technology class is obtained

from regional statistics (Opplysningsrådet for Veitrafikken, 2013).

Emissions from non-road mobile machinery in construction, industry and agriculture were originally produced by Statistics Norway, spatially distributed at the district level and thereafter gridded at 1 km resolution. The previous data stems from different years in each model domain, being in Drammen from 2012, in Oslo from 1995, in Stavanger from 1998 and in Trondheim from 2005. Non-road mobile machinery is not available in Grenland and Nedre Glomma.

For all cities except Oslo, emissions from shipping are obtained from the Norwegian Coastal Administration based upon the automatic identification system (AIS) following the methodology of (Winther et al., 2014). In the case of Oslo, emissions were estimated following a bottom-up approach based on the port activity registering system (López-Aparicio et al., 2017). This includes detailed information on arrivals, departures and operating time for individual vessels. Industrial emissions were originally provided by Statistics Norway. Industrial emissions are usually linked to the geographical position of large point

sources. However, when this was not possible, they were distributed spatially based on surrogate data, as e.g. employment figures in the industrial sector. For some locations (e.g., Grenland; Table 4), the original dataset of industrial emissions was outdated. In this case, emissions were evaluated and updated based on information from the Norwegian Pollutant Release and Transfer Register.

Table 5 describes how each sector is represented by the different possible emission types, e.g., line or area sources and also

presents the ratios between NO and $NO_2$ for the $NO_x$ emissions. The fraction of $NO_2$ in emitted $NO_x$ (as $NO_2$ mass equivalent) varies between 4.5% and 45.9% depending on the source.

The initial and background hourly concentrations used in the simulations are obtained from the CAMS regional air quality forecast production system (Marécal et al. 2015). The NetCDF files containing NO, $NO_2$, and ozone for a domain covering all of Norway and all vertical levels (0 m, 50 m, 250 m, 500 m, 1000 m, 2000 m, 3000 m, and 5000 m) came from the CAMS

online data portal: https://atmosphere.copernicus.eu/data (last access: 16 May 2019). The CAMS regional forecast data is selected for each city domain, and then interpolated horizontally and vertically to the gridding used in EPISODE. In this case study, we used the 34 vertical levels shown in Table 6. Table 6 also gives information on the size of each model domain, and the number of receptor points used.





# 4 Results and Evaluation of Model Performance

## 4.1 Mapping and Evaluation of Annual and Seasonal Model Results

### 4.1.1 Annual Mean Concentration Mapping

We now present a series of annual mean $NO_2$ concentration maps for the year 2015. One of the primary aims behind the
development of the EPISODE model was to create a system capable of mapping air pollution at high spatial resolution and at
scales relevant for human exposure within urban areas. We, therefore, apply a post-processing methodology (outlined in
Appendix A, the Pollution Mapping Post-Processing Methodology) to the irregularly spaced receptor point in order to create
the pollution maps for each city on a regular 100 m grid. Note that this post-processing method is only applied for visualisation
purposes and that for model evaluation (see Sect. 4.1.2) and exposure assessment purposes, the receptor point concentrations
($C_{rec}$) are used directly.

We present $NO_2$ concentration maps for four out of the six model domains:  Oslo (Figure 3), Drammen (Figure 4), Nedre
Glomma (Figure 5), and Grenland (Figure 6). The four domains were selected, because they represent the general features that
we see in each domain and in addition cover all of the types of spatial variability that we get in the model results. These four
cases, therefore, provide a representative sample of the whole.

We discuss the most notable features in the annually averaged concentration maps. The most notable feature of the spatial
patterns that is present in all of the $NO_2$ annual mean concentration maps is the elevated concentrations along the principal
segments of the road network and main intersections. In this way, for example, the motorway E18 is visible in the Oslo domain
(Figure 3) running in the east-west direction along the Oslo fjord, in the Drammen domain (Figure 4) running in the north-
south direction on the right-side of the map and in the Grenland domain (Figure 6) in the southeast corner of the domain. In
addition, the E6, another motorway, is visible in Oslo running north-south to the east of the fjord and in Nedre Glomma (Figure
5) running north-south on the east side of the map. Also visible are district roads like the ones to the east of Oslo (N4, N163
and N159) and the road N234 along the north of Drammensfjorden. This reflects the main source for $NO_X$ emissions in
Norwegian cities: traffic. Oslo is the capital of Norway, has the largest population and largest number of commuters. This is
reflected on the largest hotspot area (concentrations around 40 µg/m³) of the four presented maps.

Another very notable feature of elevated $NO_2$ pollution on the maps are what appear to be point source emissions. This is the
case in Oslo in the southernmost region of the domain near to the E6 and E18 junction (Figure 4) and in Drammen at UTM
coordinates $5.65 \times 10^5$ and $6.623 \times 10^6$ and $5.69 \times 10^5$, $6.622 \times 10^6$ (Figure 5). These elevated levels are in fact due to emissions
from tunnel mouths. In the Oslo domain it represents the south and north entrances of the Nøstvet Tunnel on the E6 and in
Drammen the west and east entrances of the Strømså Tunnel. The emissions from the tunnel mouths are prescribed by creating
road segments at either end with elevated traffic levels.

Oslo and Drammen are characterized by mean $NO_2$ concentrations in the sub-urban areas in the range of 10-20 µg m⁻³. Further
away, Oslo with higher emissions shows higher background concentrations with a smoother gradient from the city centre to
the forested areas with concentrations in the range 0-5 µg m⁻³. Despite Drammen having similar levels of population to the





cities in Nedre Glomma and Grenland, it still shows some relatively high $NO_2$ concentrations compared to these two less populated domains. This is because Drammen sits on the primary commuting route between Oslo and cities to the west, and thus it has a lot of extra commuting traffic.

Both the Nedre Glomma and the Grenland model domains have populations similar to the Drammen domain but divided in

two main agglomerations: Sarpsborg and Fredrikstad and Porsgrunn and Skien, respectively. This leads to NO2 annual mean concentrations in the city centres and suburban areas lower than in the other two domains shown. However, they have their particularities. In Nedre Glomma (Figure 5) the particularity is the annual average $NO_2$ concentrations of the background outside of the urban areas and away from the main roads. The background mean $NO_2$ concentration in this area does not fall below 5 µg m$^{-3}$ as it does in the Oslo and Drammen domains. This is because the rural areas in this domain are actually mostly

farmland and there are many off-road service roads that support the farmland in this area. This means there is much greater off-road activity in this area compared to the other domains, which is represented in the distribution of off-road emission sources over much of this domain.

One unique aspect of the Grenland domain is the relative prevalence of industrial pollution sources. The industry is concentrated on the Herøya peninsula at the mouth of the Posrgrunnselva river (in the centre of the domain), and on the western

side of the fjord in the southern half of the domain. Mean annual $NO_2$ concentrations are somewhat elevated in these areas with values reaching up to ~25 µg m$^{-3}$. The industrial emissions are treated as stack emissions and are injected into model layers tens of meters above the surface due to their plume buoyancy and the stack height. This explains why their impact is seen as a more diffuse zone of pollution around the industrial areas.

### 4.1.2 Full-Year and Seasonal Model Evaluation

**We evaluate the year-long NO₂ EPISODE simulations for 2015 for all six domains (Oslo, Drammen, Grenland, Nedre Glomma, Stavanger and Trondheim) outlined in Sect 3 using in-situ air quality observations of NO₂. Both the model and observation data are in hourly format and will be evaluated this way unless otherwise stated. We have in-situ observations from all six domains, but Oslo is the most well sampled city with a total of eight in-situ measurement sites compared to only two at most in the other domains. Oslo has a larger population and overall much greater emissions compared to the other cities, and it, therefore, has an increased**

**regulatory requirement to monitor its pollution. The in-situ stations that we use for the evaluation are shown in**





**Table 7. For each of the simulations in each domain, a receptor point is placed at the coordinate and height of each station shown in**

Table 7. The observations from each station are then used to evaluate these simulated the receptor point concentrations.

We first present Taylor diagrams as a means of evaluating the EPISODE model results compared to the in-situ observations. Taylor diagrams visually represent the results of three statistical tests (Pearson correlation coefficient, the root-mean-square error, and the ratio of the model standard deviation compared to the observed standard deviation) in a simultaneous fashion.

The precise manner of the representation is explained in each figure caption. The Taylor diagrams provide a good overall indication of the model performance purely from a statistical standpoint.

Figure 7 shows the results of the statistical tests for the entire year-long simulation during 2015. Looking at the σM/σO ratios, we can see that in general, the model captures the amplitude of NO$_2$ concentration variability reasonably well across all but

one of the stations (Våland) with a range in σM/σO from 0.62 to 1.40. There is neither a tendency of the model to either over or underestimate σ with almost an equal number of stations above and below 1.0. Only Våland (Stavanger) shows a high spread in modelled NO$_2$ concentrations compared to the observations with a σM/σO ratio of 1.67. We can rule out the effect of a persistent bias at Våland since the model shows only a small positive bias (+ 1.64 µg m$^{-3}$) with respect to these observations. Instead, this overestimate in the dynamic range appears to be linked to an overestimation in the NO$_2$ diurnal

variability primarily during summer (see Sect. 4.1.3 on the dynamic variability of the model). It seems possible that this occurs due to an error in the emission magnitude and variability local to Våland during summer time. The comparison with the Kannik station, also in Stavanger, supports this notion since it shows a value of σM/σO much closer to 1.0 than for Våland. All but one of the sixteen in-situ stations score values of R between 0.5 and 0.67 with only Kannik scoring lower than 0.5 at 0.49. The RMSE ranges between 0.77 and 1.18 for fifteen out of the sixteen stations. Only Våland has a much higher RMSE at 1.45,

which is linked to its much higher σM/σO ratio than the other stations. The results of each statistical test and for each station are shown in the Taylor diagrams (Figure 7, Figure 8, andFigure 9) and are summarised in Table 8. The mean values of R,



RMSE, and σM/σO for all sixteen stations are 0.6, 0.96, 1.06, respectively. This offers a means of characterising the general model performance.

We next evaluate the EPISODE model simulations only using data from the winter-time (January, February, and December combined). We carry out this specific evaluation in order to test the EPISODE model under conditions where the PSS approximation is likely fulfilled and therefore are favourable for the performance of the PSS chemistry scheme. The PSS is expected to be a reasonable approximation for conditions lacking local photochemical ozone production such as during winter in Nordic environments. Figure 8 shows the results of this evaluation in a Taylor diagram. It is very clear that evaluating the model solely during winter conditions leads to a substantial improvement in model performance scores. Now fourteen out of sixteen in-situ stations score with R values above 0.6 peaking up to 0.69. Only the stations Elgeseter (Trondheim) and Øyekast (Grenland) score below 0.6 both with values of 0.58. Excluding Våland (Stavanger), which has a σM/σO ratio of 1.42, the σM/σO ratios range between 0.54 to 1.23 for the remaining stations. Please refer to the earlier discussion of Figure 7 regarding the high modelled $NO_2$ concentration variability. Compared to the evaluation of the annual results, the winter-time results show lower values of σM/σO and general model tendency to underestimate the standard deviation of the $NO_2$ concentrations. An examination of the temporal variability (not shown) indicates that the stations with the lowest σM/σO, i.e., Manglerud, Kirkeveien, Bygdøy Alle, Hjortnes and Alnabru in Oslo, and Elgeseter in Trondheim, all tend to underestimate peak daytime $NO_2$ concentrations. The RMSE is reduced overall for the sixteen stations; the RMSE ranges between 0.74 and 1.00 with only Våland showing and an RMSE of 1.09 for similar reasons as explained earlier. The mean winter-time statistics for all sixteen stations are shown in Table 8, and these demonstrate a notable improvement in performance compared to the annual statistics. We also checked the statistics during the autumn (no figures shown), and we see an improved performance during the period September 1st to November 30th (see Table 8) relative to the rest of the year and the summer.

We present evaluation results only for the summer time in the Taylor diagram shown in Figure 9. We can see a notable degradation in model performance in terms of R and RMSE for all stations. In addition, half of the model stations now show anomalously high σM/σO ratio with values of 1.3 or above. We attribute this poorer model performance to the lack of photochemical production of $NO_2$ and ozone in the PSS chemistry scheme, without this process we should expect a different diurnal variability in $NO_2$ concentrations from that observed. Even in Oslo, we should expect at least some limited ozone production during the summer months. Table 8 shows the mean statistics for the thirteen stations shown in Figure 9, and the R and RMSE statistics show an overall degraded performance relative to the annual and winter-time evaluations.

We next evaluate the model performance using the DELTA tool target plots (Monteiro et al., 2018; Thunis and Cuvelier, 2018; Thunis et al., 2012). These plots offer a means of evaluating different aspects of model performance directly on the axes of the plots, i.e., normalised bias and the centred root mean square error (CRMSE), on the *x* and *y*-axes, respectively. The DELTA Tool plots also offer a means to evaluate the model within the context of the EC Directive while also considering the observation uncertainty. Thus, this type of evaluation offers a different perspective from the statistical measures in the Taylor diagram evaluations. Further details of the DELTA Tool method consult Appendix B and the references above. The position





of a particular model-observation pair (individual points show the results for single stations) in each quadrant tells about which type of error dominates over the other. Specifically, correlation error expressed as R dominates over standard deviation error in the left quadrants, and vice-versa in the right quadrants. Meanwhile, points in the upper quadrants indicate positive model bias and the contrary in the lower quadrants. Additionally, the tool uses the CRMSE and normalised bias to calculate a target

value, which is also visualised on the target plot as the distance from the origin. The objective is to have points with a target value of 1 or less and thus lie within the green area of the plots.

Separately, the DELTA tool calculates the model quality indicator (MQI) (see Appendix B and enclosed references for further details), which determines whether the model-observation bias is less than the observation uncertainty. Furthermore, Monteiro et al. (2018) and Thunis and Cuvelier (2018) define the model quality objective (MQO) as to whether the 90[th] percentile of

MQI for all stations is less than 1. If this criterion is satisfied the model quality objective is considered to be satisfied.

Figure 10Figure 11, and Figure 12 show the target plots for the year-round evaluation, the winter-time only evaluation, and the summer-time only evaluation, respectively. The target plots highlight an important and consistent feature of the model performance throughout the year, which is that all of the model-observation evaluation pairs lie in the upper and lower left-hand quadrants. This indicates the correlation error, expressed as R, dominates the contribution to the CRMSE error term.

We first discuss the annual evaluation shown in Figure 10. The MQO is satisfied for the annual evaluation with the 90[th] percentile of the MQI calculated at 0.971. Only one station, Bygdøy Alle, has a large enough negative bias to have a specific MQI value above 1. Overall, an equal number of stations have a positive and negative normalised bias. However, there is an apparent signal in the Oslo results for a negative bias in this evaluation, and the magnitude of the negative biases is slightly larger than the positive biases.

EPISODE just achieves the MQO during winter with the 90[th] percentile of MQI being calculated at 0.995. This is despite two stations in Oslo, Åkebergveien and Bygdøy Alle, showing larger than acceptable low biases during the winter-time evaluation period. Both stations are visible in the lower left quadrant outside of the green target zone. Given the reasonable correlations at both stations, we can perhaps infer that a persistent model or emission process is the cause of this effect. Further study will be required to determine in detail the cause of this. Looking at the overall bias evaluation in the target plot, we can see that

there are now more stations with a negative bias than a positive bias. Similarly as in Figure 10, the magnitude of the negative bias is larger than the magnitude of the positive bias, and, similarly, it is the Oslo stations that have a greater tendency to show a negative bias. We note that the MQO is also satisfied during the autumn (figure not shown) with the 90[th] percentile of the MQI being calculated at 0.996.

Despite the degraded performance shown in the Taylor diagram for the summer in Figure 9, the MQO is satisfied for the

summer time analysis with the 90[th] percentile MQI being calculated at 0.933. Please note the exclusion of the Kannik and Våland stations. Following guidelines from the EC air quality Directive, neither had sufficient observations during the summer to be able to perform the DELTA tool target plot analysis. Despite this limitation, we can show that the MQO is satisfied across the thirteen remaining stations indicating that overall the model bias is sufficiently and consistently low enough at these locations on an hour by hour basis. The overall bias statistics show no strong prevalence for the number of stations with a



negative or positive bias, but, as before, the magnitude of the negative bias is ever so slightly larger and we see the negative bias affecting the Oslo stations in preference to the other cities.

### 4.1.3 Evaluation of the Dynamic Behaviour of EPISODE

We next evaluate the temporal variations in $NO_2$ concentrations simulated by EPISODE. We do this by looking at the
difference in mean observed and simulated $NO_2$ concentration between the summer and winter, between night and day, and between the weekdays and the weekend. This evaluation examines various processes within the model including the photochemistry arising from changing sunlight and background ozone, the temporal variability in emissions, and changes in the meteorology impacting the mixing volume. In addition, seasonal changes in background ozone concentrations impact the equilibrium between NO and $NO_2$ in a complex manner throughout the year.

We first look at the differences between mean summer and winter $NO_2$ concentrations (henceforth referred to as Δsummer-winter) shown in Figure 13. Based on the reduced sunlight and higher frequency of inversion events leading to reduced mixing volumes in the boundary layer during winter, on average we should expect higher winter-time $NO_2$ concentrations compared to those during summer. Indeed, we can see in Figure 13 that this is the observed behaviour since all of the model-observation pairs occupy the left half of the plot areas. Further, the model reproduces this general behaviour in every instance. Looking in
more detail we can see that there is a tendency of the model to underestimate Δsummer-winter. Due to the sign of the difference (negative), this could occur due to a model underestimate of the winter $NO_2$ concentrations or an overestimate by the model during summer. Looking at the model biases for each season (not shown), we can determine that this occurs due to a combination of effects. Large underestimates of winter-time $NO_2$ concentrations seem to account for the lowest instances of Δsummer-winter at Kirkeveien, Manglerud, Bygdøy Alle and Hjortnes in Oslo, and Elgeseter in Trondheim. We attribute this
error either to an underestimate in the winter-time emissions at these stations or due to the effects of an underestimate in the frequency or intensity of inversion events. Meanwhile, overestimates of summer time $NO_2$ concentrations seem to account for lower Δsummer-winter at RV4 Aker Sykehus (Oslo), Bangeløkka (Drammen), and St Croix (Nedre Glomma). Various possible explanations of this overestimate exist, e.g., a high bias in the background summer ozone estimates, an overestimate of the summer time $NO_x$ emissions at these stations, or an underestimate of the summer time $NO_2$ photolysis rates. We lack the
observations to determine which hypothesis best explains this error and further study will be required to understand this aspect of the model behaviour. However, given the results in Figure 14 showing that the model captures day minus night differences fairly consistently without a systematic error, a persistent error in the $NO_2$ photolysis rates seems a less likely explanation than either a problem with the emissions or the background ozone.

We next evaluate the model performance in its ability to capture the mean day minus night differences in $NO_2$ concentrations
(henceforth referred to as Δday-night). The model is able to generally capture the observed behaviour, i.e., higher $NO_2$ concentrations during the day compared to the night. This general behaviour arises from the fact that most activity associated with the emissions of $NO_x$ occurs during the day. Looking in detail we can see that most of the points lie close to the dashed





line indicating that, on the whole, the model captures this property very well. There are five points that lie further below the dashed line than most with lower modelled Δday-night than was observed. An examination of the day and night model biases for four (Elgeseter, Bygdøy Alle, Manglerud, and Kirkeveien) of these locations (not shown) shows that the underestimation of Δday-night is driven by a low bias in the modelled mean day time $NO_2$ concentration. However, the model also

underestimated the night-time $NO_2$ concentrations but to a much smaller degree. The fact that the day and night-time biases share the same sign, but the day time biases are much larger during the period of largest emission, gives a strong indication that underestimates in the overall magnitude emissions are responsible for these biases at these locations. Further study will be required to determine the exact nature of this discrepancy. In the case of the fifth location, Våland, showing a lower modelled Δday-night than was observed, we find a strong seasonality whereby large errors in Δday-night manifest themselves during

summer with night-time NO2 concentrations regularly exceeding day-time values. Since this effect is not found at Kannik, which is located within 1 km of Våland, we believe this error to be caused by errors in the diurnal variability of the NO2 emissions local to Våland. Further study will be required to identify definitively the cause of this error.

Lastly, we look at the difference between mean weekday and mean weekend $NO_2$ concentration in Figure 15 (henceforth referred to as Δweekday-weekend). As we would expect, there are higher $NO_2$ concentrations observed during the weekdays

compared to the weekend due to the higher levels of activity during the week. This aspect of the model behaviour is linked entirely to the temporal variability applied to the emissions such that weekdays have higher emissions, and also different diurnal variability in the emissions. Owing to the specification of the weekly variability in the emissions the model reproduces this behaviour in a general way very well with most of the station points lying on the 1:1 line within 5 µg m$^{-3}$ of the observed Δweekday-weekend. Only RV4 Aker Sykehus in Oslo is slightly higher than this level with a difference of 6.5 µg m$^{-3}$, and this

seems to be linked to a changing model bias from the weekdays to the weekend whereby the bias decreases at the weekend. In the case of the other stations, the bias during the week and weekend remains relatively consistent. We attribute the larger error at RV4 Aker Sykehus to an error in the temporal variability at this location since over an entire year there is no other physical process that can account for this. Further study will be required to understand this in more detail. This gives a good example of how EPISODE can potentially be used to improve our understanding of emissions at the fine scale in a city.

**4.2 The Model's Capability of Capturing Pollution Episodes**

We now present examples of the ability of EPISODE to map $NO_2$ pollution during pollution events. Following an examination of the observed time series of $NO_2$ pollution for Oslo and Drammen, we select two periods of interest (one for each city) where $NO_2$ concentrations became elevated over a few days. The first is an event that took place in Oslo between December 9th and 13th, and the second took place in Drammen between January 4th and 7th. We analyse each event using daily mean maps from

the worst day of the pollution event, time series for selected stations, and statistical scores for all of the available stations.





### 4.2.1 Oslo Pollution Episode 9th-13th December 2015

Figure 16 shows a map of the daily mean $NO_2$ concentrations over the Oslo domain for December 11th. The map shows significant elevated $NO_2$ concentrations over large areas of the domain. Average levels of 40 µg m$^{-3}$ and higher were present over most of the urban areas in and around the city with levels of 60 µg m$^{-3}$ and higher present in the central and eastern areas

of Oslo and along major roads outside the city. On December 11th the meteorological condition involved winds from the south, and the effects of this are clearly visible in the form of plumes to the north of roads running east-west to the east of Oslo. Light southerly winds persisted throughout the event.

Figure 17 is a plot of the time series for the observed and modelled $NO_2$ concentrations at two measuring sites in Oslo, Åkebergveien and Manglerud. These two stations are selected because they exhibit different characteristics of the pollution

episode with different timings for the onset, and the model exhibits different performance statistics for each station (see Table 9). Chronologically, the peaks in pollution above 60 µg m$^{-3}$ occur first at Manglerud (and other stations in the west of Oslo) on December 10th before the pollution also peaked above this level on December 11th at Åkebergveien. The model is able to capture this difference at both stations. Further, the model captures the shorter duration of the peak in $NO_2$ concentrations at Manglerud on December 10th compared to the other days.

We can see that the σM/σO ratio is lower than 1.0 for all of the comparisons against in-situ stations. Looking at Figure 17 we can see that the model captures the night-time minima reasonably well, but underestimates peak $NO_2$ concentrations. This underestimate of the peak is either due to uncertainties in boundary layer meteorology, the emissions magnitude or time variability of the emissions. Further study will be required to determine the exact cause. Overall, these scores demonstrate acceptable model performance during this pollution episode. This highlights the ability of the EPISODE model to capture

individual pollution events when coupled with meteorological forcing and background pollutant concentrations of sufficient quality.

### 4.2.2 Drammen Pollution Episode 4th-7th January 2015

Figure 18 shows a map of the mean $NO_2$ concentrations over the Drammen domain for January 5th. During the four-day pollution episode, the worst pollution occurred on January 5th, which is why we selected this date to visualise. EPISODE

simulates concentrations of 30 µg m$^{-3}$ and higher over much of the populated areas in and around Drammen along the Drammenselva river and to the North and South of the fjord along the E18 highway. Only the settlement of Konnerud (in the south-central area of the domain) avoided levels over 30 µg m$^{-3}$. Low wind speeds with no distinct wind pattern were present on January 5th. In the days preceding the 5th there were light winds from the west, and following it there was no clear wind pattern and only very low wind speeds until January 7th.

We display both the observed and modelled receptor point $NO_2$ concentrations for Bangeløkka in Figure 19. We can see the onset of the event starting on the afternoon of January 4th that then persisted for three more days each with peak $NO_2$ concentrations over 60 µg m$^{-3}$. After January 4th, the peak in $NO_2$ concentrations began from 6 am onwards peaking between



9 am and 11 am. A smaller secondary peak occurs in the late afternoon associated with the evening rush hour. We evaluated the model performance during this period of time against the Bangeløkka in-situ station in Drammen. The ratio of the modelled and observed standard deviation, $\sigma M/\sigma O$, was 0.82, indicating lower than observed model variability. The Pearson correlation, R, was 0.8, the RMSE was equal to 0.59, and the mean bias was +4.3 µg m$^{-3}$. The $\sigma M/\sigma O$ is lower than 1.0 and this indicates

that the model underestimates the dynamic range of variability. Looking at Figure 19 it is possible to see that the model captures the extent of the nighttime minimum in $NO_2$ concentrations (0/24h to 6h) on two out of four occasions, but does not capture the full extent the maximum on three out of four days. Unfortunately, we only have one in-situ station available to evaluate the model in the Drammen domain for this pollution episode, which prevents a wider evaluation of the model for this episode. Within this limitation, the statistics show acceptable model performance at one of the most polluted sites in Drammen. This

again highlights that when coupled with good quality meteorological forcing and background concentrations of pollutants EPISODE can capture individual pollution events well.

## 5 Summary

The EPISODE urban dispersion model was presented, which serves as the base model for the EPISODE-CityChem extension described in Part 2 of this paper (Karl et al., 2019). EPISODE combines a 1 km 3D Eulerian grid with sub-grid scale dispersion

from point and line sources to receptor points. This allows EPISODE to provide a finer scale and higher resolution representation of pollution in urban environments than traditional regional chemistry transport models. It thus addresses one of the main weaknesses of regional air quality models, i.e., the recurring problem of representing a diverse range of urban environments (from street-side to urban background) all within a 10+ km scale grid. We presented here the simulation of $NO_2$ pollution with the EPISODE model at high resolution using a photostationary steady state chemistry scheme. EPISODE was

designed to simulate $NO_2$ pollution in Nordic low-light environments where the usage of the photostationary steady state was considered appropriate. Note that the EPISODE-CityChem extension in Part 2 includes a more comprehensive chemistry scheme suitable for a wider range of environments. We demonstrate the application of the model in six case studies in Norwegian cities for the entire year of 2015. We evaluated the model against in-situ observations of $NO_2$ concentrations in all six cities, and present more traditional statistical metrics including RMSE, R, and $\sigma M/\sigma O$ (the ratio of simulated and observed

standard deviations), and dedicated metrics for evaluating air quality models, e.g., target plot analysis and a model quality objective (Monteiro et al. 2018; Thunis and Cuvelier 2018; Thunis et al., 2012). The model satisfies the model quality objective for every time period it was evaluated for (annual, winter, autumn, and summer), and only two stations out of sixteen failed the target plot analysis. The statistics over the whole year demonstrate an overall reasonable performance throughout the year. However, more in-depth analysis of the model performance during the different seasons demonstrates significantly improved

performance, both in terms of correlation and RMSE, during autumn and winter compared to summer. This is consistent with the expectation that the photostationary steady state chemistry should perform well during the darker months of the year, and it demonstrates the suitability of EPISODE for studying the $NO_2$ pollution problem in Norway since the most elevated $NO_2$



pollution levels occur during autumn and winter. We conclude that EPISODE is suitable both for scientific study of $NO_2$ air pollution and also to support policy applications, e.g., pollution episode analysis, seasonal statistics, policy and planning support, and air quality management.

## 6 Future Work

We outline several developments that are planned in the near future aimed at improving the representation of $NO_2$ in EPISODE simulations. The first is to simulate the entrainment of ozone within $NO_x$-rich plumes from traffic emissions. Currently, the photostationary steady state is solved at each receptor point using the NO and $NO_2$ transported from the pollution sources and the grid ozone. We propose replacing the current treatment of ozone and include a simulation of ozone mixing into the $NO_x$-rich plumes linked to the stability conditions.

Another weakness of the PSS is that it solves the chemistry to equilibrium instantaneously regardless of the distance of a receptor point from a pollution source. When in reality, the equilibrium between $NO_2$, NO, and ozone may take minutes to achieve. On the short transport timescales of only tens of meters from a pollution source, this may be problematic and a treatment of the chemistry taking into account the time to reach equilibrium and the transport distance may be more appropriate. We therefore plan to develop a modification to the photostationary steady state calculations to account for this type of situation.

Lastly, we plan to introduce another modification to the photostationary steady state that will simulate the formation of $N_2O_5$, which is an important winter-time sink for $NO_x$ (Dentener and Crutzen, 1993). This will require the introduction of the chemical species $NO_3$ and $N_2O_5$ itself into the photostationary steady state scheme. $N_2O_5$ loss onto aerosols will be taken into account via an uptake coefficient onto a dynamically calculated particulate matter surface area derived from the simulation of particulate matter concentrations.

It is already possible to simulate particulate matter concentrations for $PM_{2.5}$ and $PM_{10}$ with the EPISODE model (in separate simulations from the $NO_2$ runs), but we chose not to present case studies for these pollutants in this paper. Compared to $NO_2$, the model uncertainties for simulating $PM_{2.5}$ and $PM_{10}$ are linked much more to emission processes, i.e., wood burning and road dust resuspension, respectively. Both emission processes require dedicated models external to EPISODE to estimate realistic emissions, which are beyond the scope of this paper. Running without the inclusion of these emission processes results in significantly degraded model performance compared to the $NO_2$ simulations. The standalone emission models are the

MEDVED model for wood burning emissions (Grythe et al., 2019), and the NORTRIP model for road dust resuspension (Denby et al. 2013). The offline coupling of both emission models into EPISODE for PM simulations is planned for the near future and will greatly enhance the model's capability for simulating particulate matter pollution. In addition to this, a standalone traffic exhaust emission model is being developed that will replace many of the functionalities of the AirQUIS

system.





*Code and data availability:* The source code for the EPISODE model version 10 is available under the RPL 1.5 license at https://doi.org/10.5281/zenodo.3244056. The model compilation requires installation of the gcc/gfortran fortran90 compiler (version 4.4. or later) and the netCDF library (version 3.6.0 or later).

Model input datasets are available from the NILU ftp server upon request. These datasets include: meteorological, emission,
and ancillary input files for the entire year 2015; output model data for all of the 2015 simulations; and data in the format for the DELTA Tool analysis package.

**Appendix A: Pollution Mapping Post-Processing Methodology**

The visualisation in the maps is created by first subtracting from each receptor point concentration, $C_{rec}$, the Eulerian grid
concentration, $C_m$, for the corresponding grid square in which the receptor point resides following,

$C_{local} = C_{rec} - C_m$     (A1)

which leaves the local concentration residual, $C_{local}$. Next, the Eulerian grid concentration field at 1 km resolution, $C_m$, is interpolated to the coordinates, $(x^r, y^r, 1)$, of each receptor point using a spline method to give, $C_{m,rec}$, following

$C_{m,rec} = F_{int}(C_m, [x^r, y^r])$     (A2)

Then both the residual from Eq. (A1) and $C_{m,rec}$ are added together to determine the receptor point concentration, $C_{rec*}$, which now contains both the receptor point and the interpolated Eulerian grid components. Finally, the modified concentrations for all of the irregularly spaced receptor points, $C_{rec*}$, are then re-gridded onto a 100 x 100 m grid covering the entire domain using tri-linear interpolation.

In practice, there are many areas within the urban centre with receptor point sampling at higher spatial resolutions than 100 m.
Thus, 100 m represents a conservative choice for the effective mapping resolution in these important areas. This post-processing step also serves to remove the visual imprint of the 1 x 1 km Eulerian grid (remember that receptor point concentrations are a sum of the Eulerian grid and local contribution following Eq. (7) from the gridded receptor point concentrations.

**Appendix B: Statistical indicators and model performance indicators**

The model is evaluated with the following statistical metrics: the ratio of the modelled and observed standard deviation ($\sigma M / \sigma O$), root mean squared error (RMSE), centred root mean square error (CRMSE), Pearson's correlation coefficient (R), normalised mean bias (NMB), and index of agreement (IOA).

The respective standard deviations of the model and observations are calculated via

$$\sigma M = \sqrt{\frac{1}{N} \sum_{i=1}^{N}(M_i - \bar{M})^2}$$     (B1)

$$\sigma O = \sqrt{\frac{1}{N-1} \sum_{i=1}^{N}(O_i - \bar{O})^2}$$     (B2)

The RMSE provides a representation of the magnitude of the error for each hourly model-observation pair and is defined as:


$$RMSE = \sqrt{\frac{1}{N}\sum_{i=1}^{N}(M_i - O_i)^2} \qquad (B3)$$

$$R = \frac{\frac{1}{N}\sum_{i=1}^{N}(M_i - \bar{M})(O_i - \bar{O})}{\sigma M \ \sigma O} \qquad (B4)$$

The IOA is defined as

$$IOA = 1 - \frac{\sum_{i=1}^{N}(M_i - O_i)^2}{\sum_{i=1}^{N}(|M_i - \bar{M}| + |O_i - \bar{O}|)^2} \qquad (B5)$$

When the IOA is equal to 1 it indicates perfect agreement between the model and observations and a value of zero indicates no agreement at all.

The CRMSE and normalised mean bias are used in the axes of the DELTA Tool target plots and are calculated as follows:

$$CRMSE = \sqrt{\frac{1}{N}\sum_{i=1}^{N}((O_i - \bar{O}) - (M_i - \bar{M}))^2} \qquad (B6)$$

$$NMB = \frac{\bar{M} - \bar{O}}{\bar{O}} \qquad (B7)$$

In addition to these metrics, we also evaluate the model according to the DELTA Tool model quality indicator (MQI) and the related model quality objective (MQO) (Monteiro et al., 2018; Thunis and Cuvelier, 2018). The MQI calculation provides an advanced evaluation of model performance by considering the observation uncertainty on each individual measurement, $U_{95}(O_i)$, which is defined as:

$$U_{95}(O_i) = k\ u_r^{RV}\sqrt{(1 - \alpha^2)O_i^2 + \alpha^2(RV)^2} \qquad (B8)$$

Where $u_r^{RV}$ is the relative measurement uncertainty estimated around a reference value, RV, for a given time averaging, e.g., hourly or daily limit values of the air quality directive. $\alpha^2$ is the fraction of the uncertainty around RV, which is non-proportional to the concentration level, and k is the coverage factor that scales the error in order to achieve a specific confidence interval. k is most typically set to 2 in order to achieve a 95% confidence interval.

The root mean square of the observation error is calculated via:

$$RMS_U = \sqrt{\frac{1}{N}\sum_{i=1}^{N}(U_{95}(O_i))^2} \qquad (B9)$$

The MQI is then define as the ratio between the absolute model-observation bias and a quantity proportional to the observation uncertainty via:

$$MQI = \frac{|O_i - M_i|}{\beta\ RMS_U} \qquad (B10)$$

Where β is a scaling set to 2 in the DELTA Tool. In the DELTA Tool target plots, MQI is the distance between the origin and a point on the plot for a given station. The MQO is considered fulfilled when MQI ≤ 1. Following the Air Quality Directive requirements, the DELTA Tool sets a criteria whereby the MQO is defined as being satisfied when the MQI is fulfilled for at least 90% of the stations. In other words, ranking the station MQIs in ascending order, the inferred 90[th] percentile must be 1 or lower.





*Author contributions*.

**PDH**: wrote the main text of the paper; developed a technical description of the model based on an in-depth evaluation of the code; developed the scientific questions; ran the model case studies for the six Norwegian cities; analysed and evaluated the

EPISODE model results presented in this work.

**SEW**: implemented the HIGHWAY-2 line source dispersion model and the two point source dispersion models in EPISODE.

**GSS**: Supported scientific design of the six city case studies and made significant developments to EPISODE code, e.g., development of coupling with AROME meteorological data.

**MV**: post-processed the EPISODE model results and visualized the results as the maps.

**DVT**: prepared the emissions used in the EPISODE model runs.

**SAL**: provided the technical and scientific guidance for the preparation of the emissions used in the EPISODE model runs.

**MOPR**: testing of the UECT and TAPM4CC pre-processing utilities, assisted with the DELTA Tool.

**MK**: prepared the observation and modelling data into the correct formats for the DELTA Tool; wrote the technical supplements and made contributions to the main text.

*Competing interests*. The authors declare that they have no competing interests.

*Acknowledgements*. The authors thank NILU for internal funding used to support this work. NILU thanks Leif Håvard Slørdal (retired) for his major contributions to the development of EPISODE. LHS declined co-authorship, but EPISODE exists today due his dedicated work. PDH wishes to thank Virginie Marécal for her scientific discussion in support of this article.

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

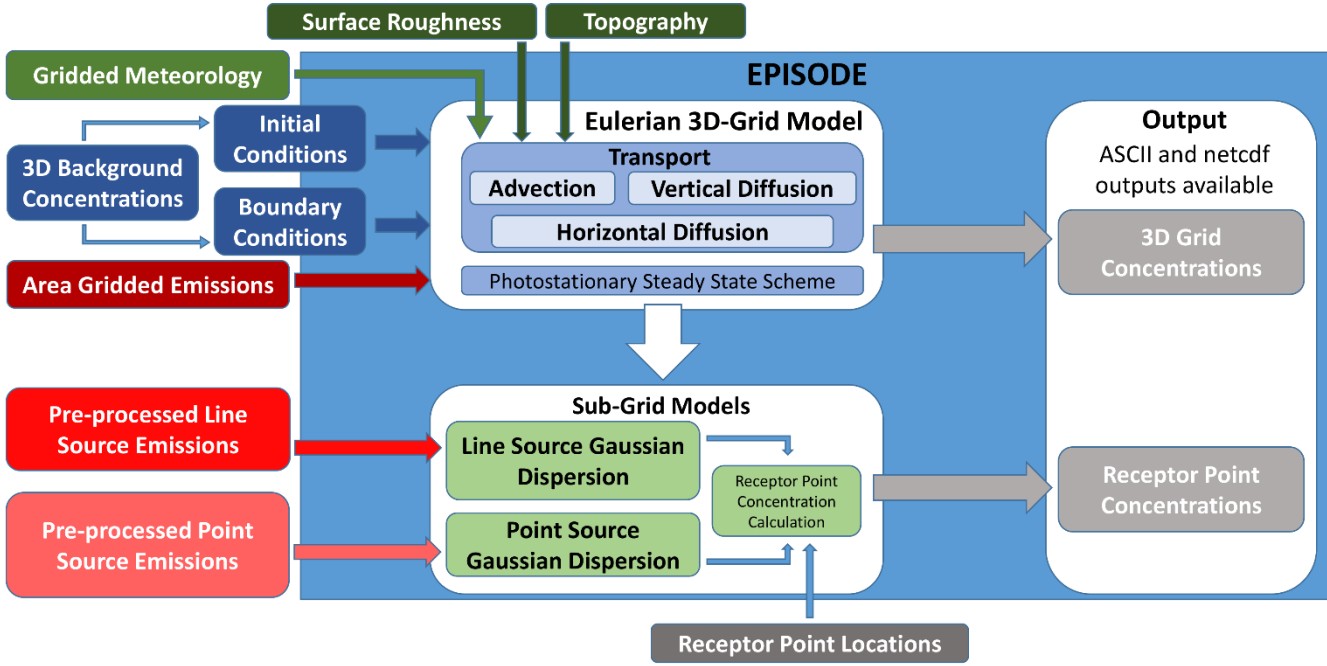

**Figure 1. Schematic diagram of the EPISODE model. The large blue box represents operations carried out during the execution of the EPISODE model. The components of the EPISODE model are the Eulerian grid model and the sub-grid models. The inputs for EPISODE are specified on the periphery.**





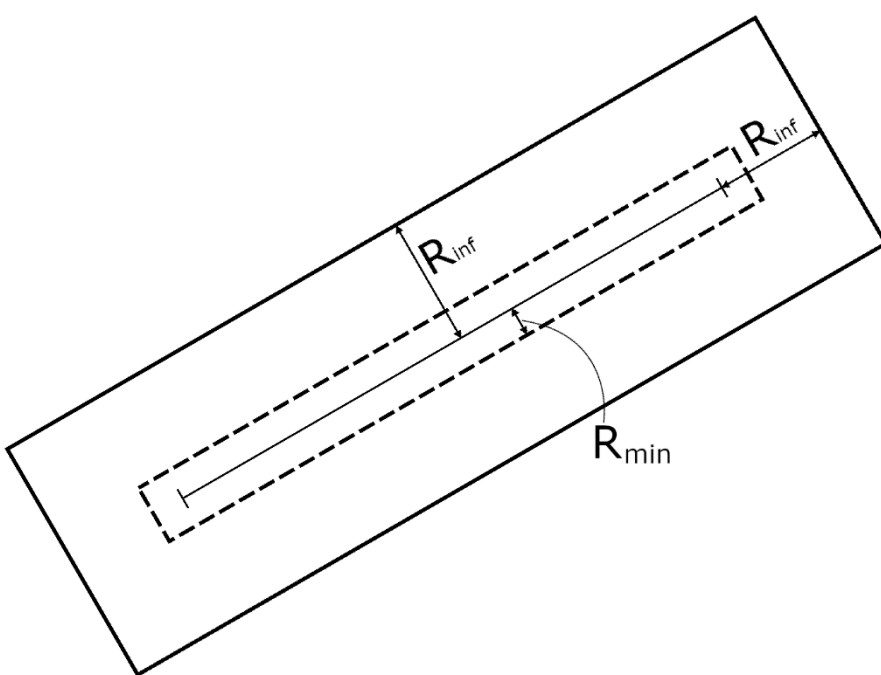

**Figure 2. An illustration of the rectangular area of influence around an example road link showing the minimum ($R_{min}$) and maximum ($R_{inf}$) distances influenced by a line source.**


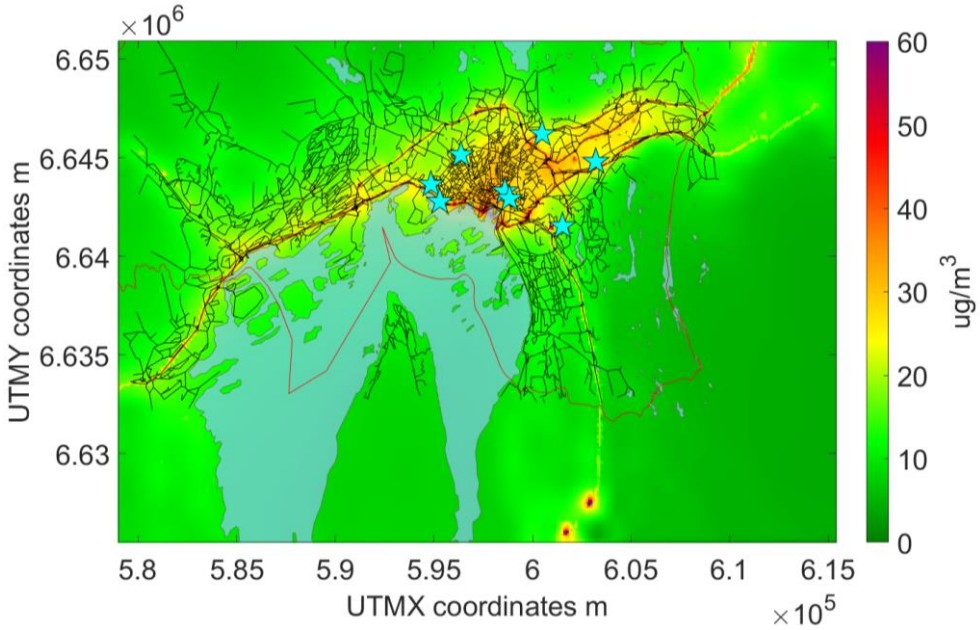

**Figure 3. Annually averaged NO₂ concentrations (μg/m³) from the EPISODE model over the Oslo domain at 100 m x 100 m horizontal resolution. The concentrations are derived from the receptor point concentrations and then re-gridded onto a 100 m grid. The colour scale shows the range in annual mean NO₂ concentrations between 0 and 60 μg m⁻³. The light blue stars indicate the locations of the air quality observation stations (Table 7). The blue areas represent the sea, lakes and rivers. The red lines represent administrative boundaries, and the black lines are roads. © OpenStreetMap contributors 2019. Distributed under a Creative Commons BY-SA License.**



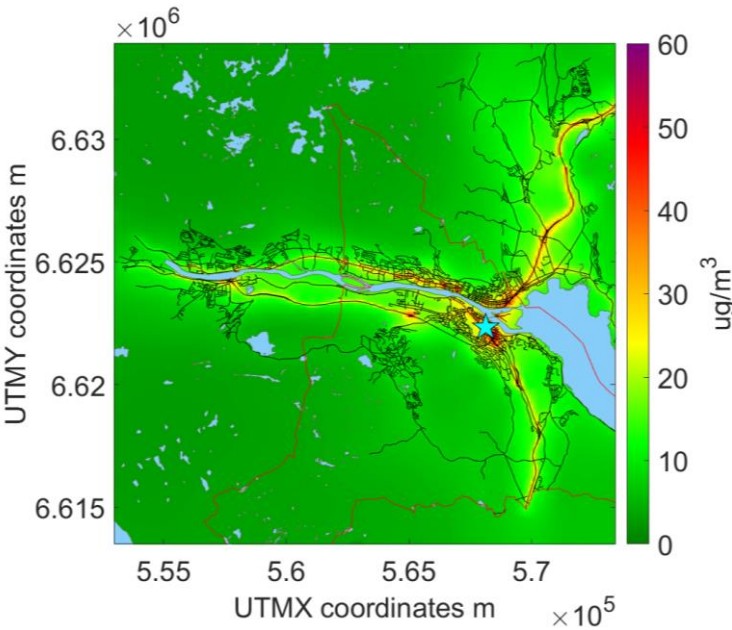

**Figure 4. Annually averaged NO$_2$ concentrations (µg/m$^3$) from the EPISODE model over the Drammen domain at 100 m x 100 m horizontal resolution. The concentrations are derived from the receptor point concentrations and then re-gridded onto a 100 m grid. The colour scale shows the range in annual mean NO$_2$ concentrations between 0 and 60 µg m$^{-3}$. The light blue stars indicate the locations of the air quality observation station (Table 7). The blue areas represent the sea, lakes and rivers. The red lines represent administrative boundaries, and the black lines are roads. © OpenStreetMap contributors 2019. Distributed under a Creative Commons BY-SA License.**



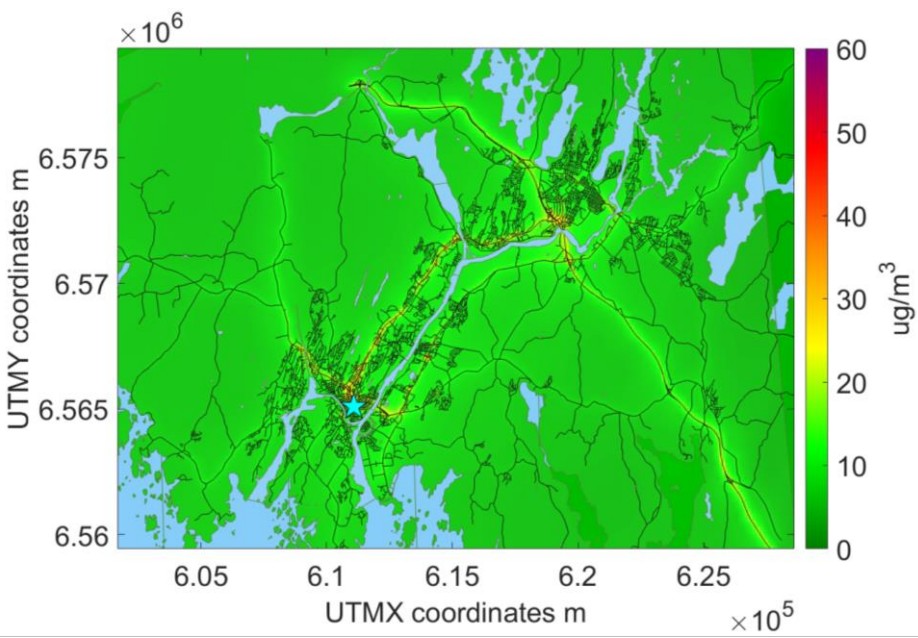

**Figure 5. Annually averaged NO$_2$ concentrations (µg/m$^3$) from the EPISODE model over the Nedre Glomma domain at 100 m x 100 m horizontal resolution. The concentrations are derived from the receptor point concentrations and then re-gridded onto a 100 m grid. The colour scale shows the range in annual mean NO$_2$ concentrations between 0 and 60 µg m$^{-3}$. The light blue stars indicate the locations of the air quality observation station (Table 7). The blue areas represent the sea, lakes and rivers. The red lines represent administrative boundaries, and the black lines are roads. © OpenStreetMap contributors 2019. Distributed under a Creative Commons BY-SA License.**



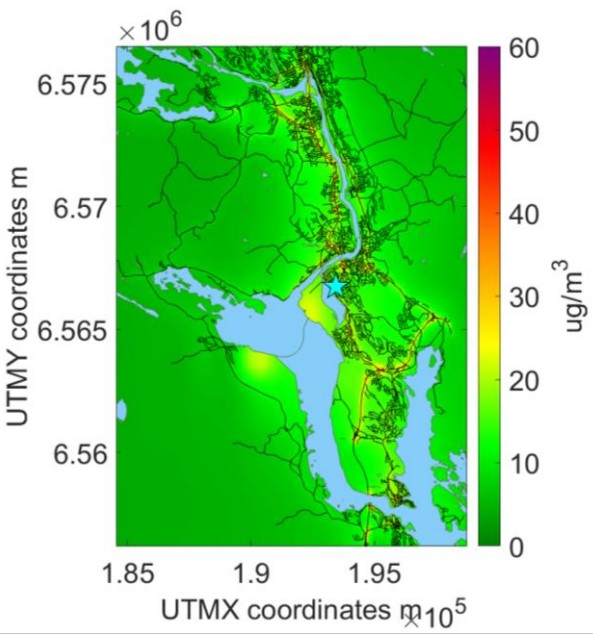

**Figure 6. Annually averaged NO₂ concentrations (μg/m³) from the EPISODE model over the Grenland domain at 100 m x 100 m horizontal resolution. The concentrations are derived from the receptor point concentrations and then re-gridded onto a 100 m grid. The colour scale shows the range in annual mean NO₂ concentrations between 0 and 60 μg m⁻³. The light blue stars indicate the locations of the air quality observation station (Table 7). The blue areas represent the sea, lakes and rivers. The red lines represent administrative boundaries, and the black lines are roads. © OpenStreetMap contributors 2019. Distributed under a Creative Commons BY-SA License.**



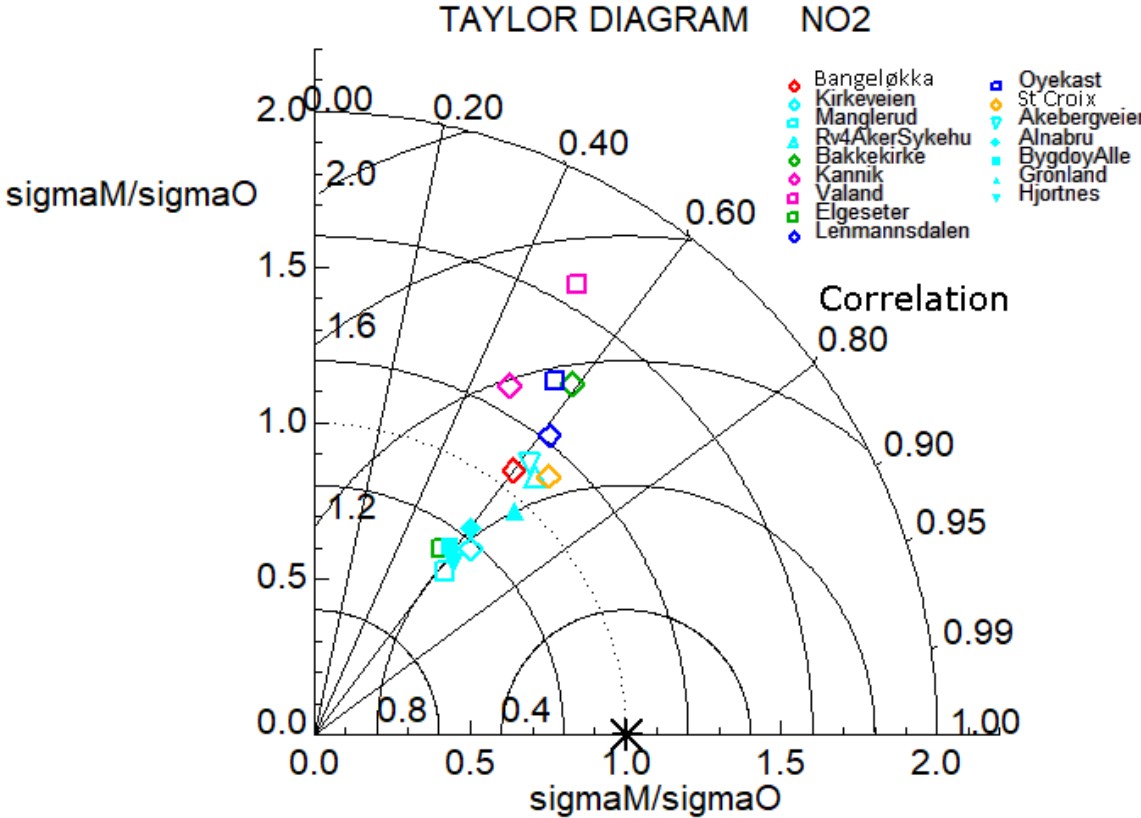

**Figure 7. A Taylor diagram calculated using the annual hourly time series of NO₂ concentrations for all sixteen in-situ stations used for the model evaluation across all six domains. The symbols are colour coded according to each model domain where Drammen is red, Oslo is cyan, Trondheim is green, Stavanger is pink, Grenland is dark blue, and Nedre Glomma is orange. The x and y-axis both represent the ratio of the model standard deviation to the observed standard deviation in NO₂ concentrations for a particular station, such that points can be plotted on concentric circles centred on the x/y origin. The correlation is plotted according to the azimuthal angle from the origin represented as a series of straight lines emanating from the x/y origin. Lastly, the RMSE is also represented for each station according to their linear distance from 1.0 on the x-axis.**



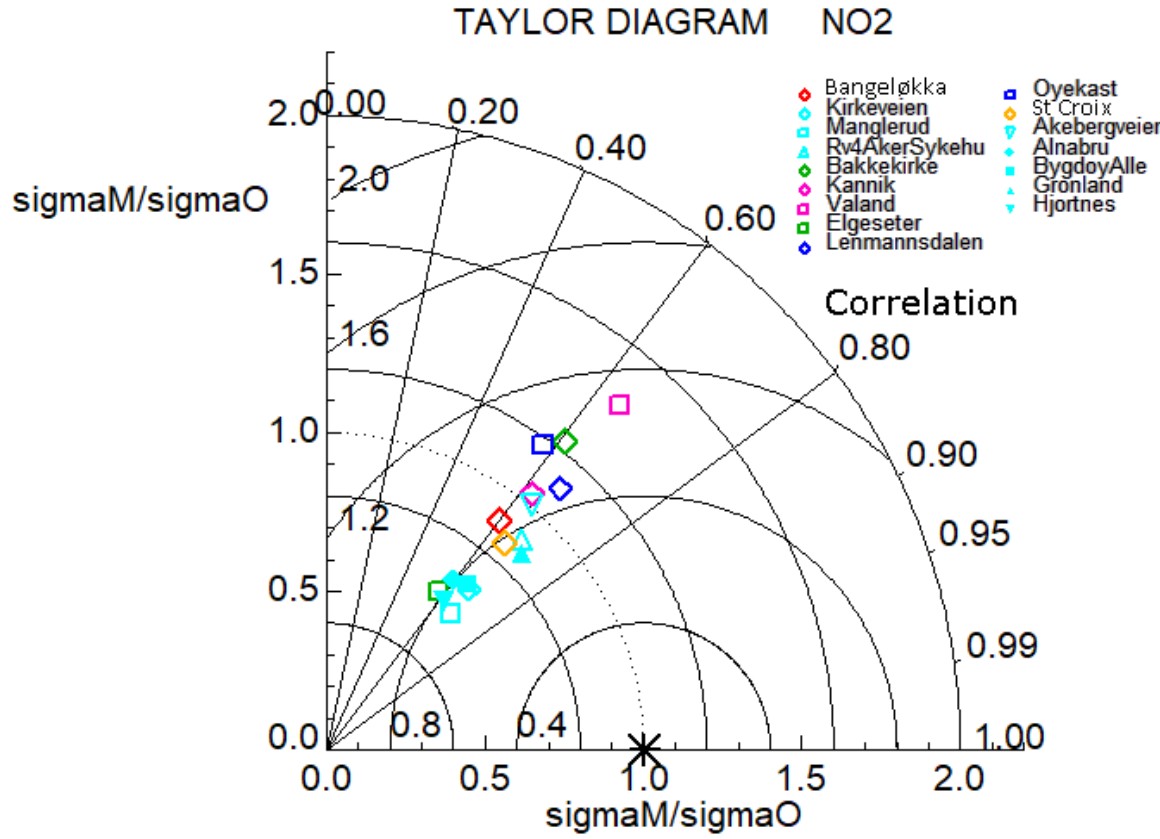

**Figure 8. A Taylor diagram calculated using the winter only (January, February, and December) hourly time series of NO₂ concentrations for all sixteen in-situ stations used for the model evaluation across all six domains. The symbols are colour coded according to each model domain where Drammen is red, Oslo is cyan, Trondheim is green, Stavanger is pink, Grenland is dark blue, and Nedre Glomma is orange. The x and y-axis both represent the ratio of the model standard deviation to the observed standard deviation in NO₂ concentrations for a particular station, such that points can be plotted on concentric circles centred on the x/y origin. The correlation is plotted according to the azimuthal angle from the origin represented as a series of straight lines emanating from the x/y origin. Lastly, the RMSE is also represented for each station according to their linear distance from 1.0 on the x-axis.**



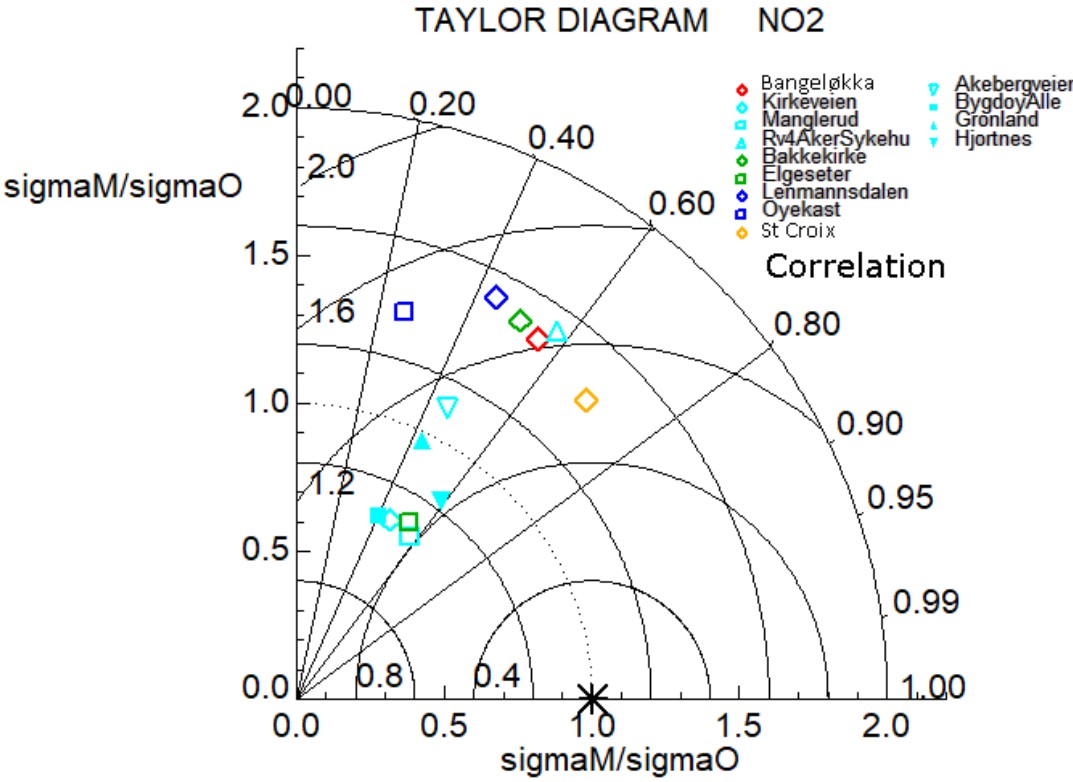

**Figure 9. A Taylor diagram calculated using the summer only (June, July, and August) hourly time series of NO₂ concentrations for thirteen in-situ stations used for the model evaluation across five out of the six domains (excluding Stavanger). The symbols are colour coded according to each model domain where Drammen is red, Oslo is cyan, Trondheim is green, Stavanger is pink, Grenland is dark blue, and Nedre Glomma is orange. The x and y-axis both represent the ratio of the model standard deviation to the observed standard deviation in NO₂ concentrations for a particular station, such that points can be plotted on concentric circles centred on the x/y origin. The correlation is plotted according to the azimuthal angle from the origin represented as a series of straight lines emanating from the x/y origin. Lastly, the RMSE is also represented for each station according to their linear distance from 1.0 on the x-axis.**





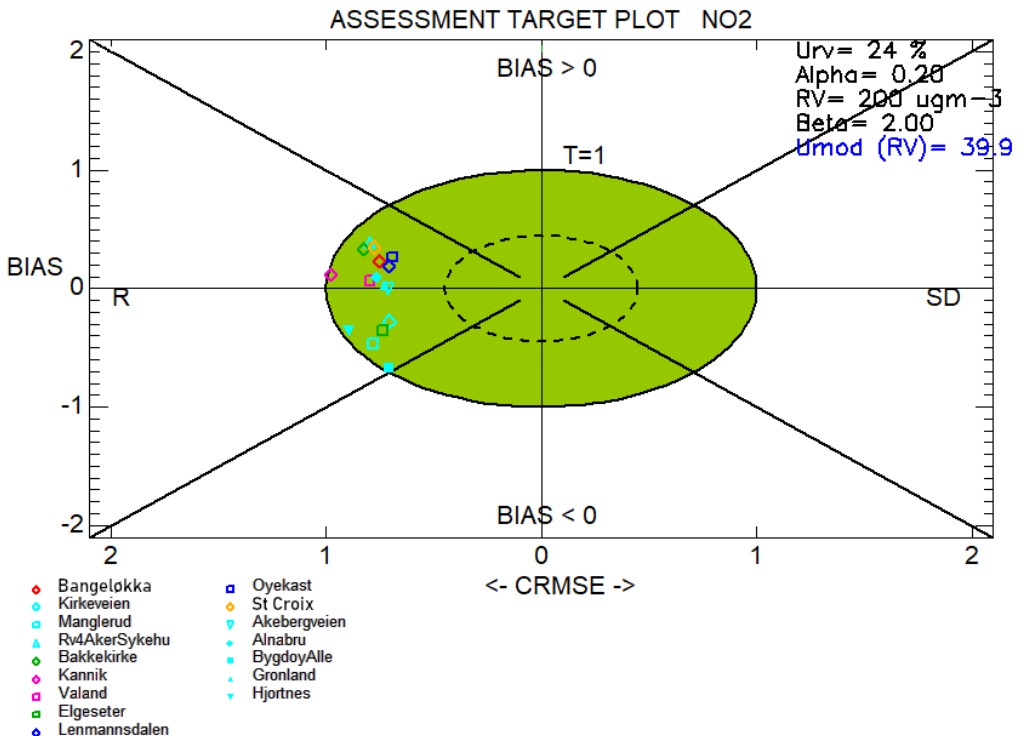

**Figure 10. Target plots created with hourly time series of NO₂ concentrations for 2015 for all sixteen in-situ stations used for the model evaluation across all six domains. The symbols are colour coded according to each model domain where Drammen is red, Oslo is cyan, Trondheim is green, Stavanger is pink, Grenland is dark blue, and Nedre Glomma is orange.**



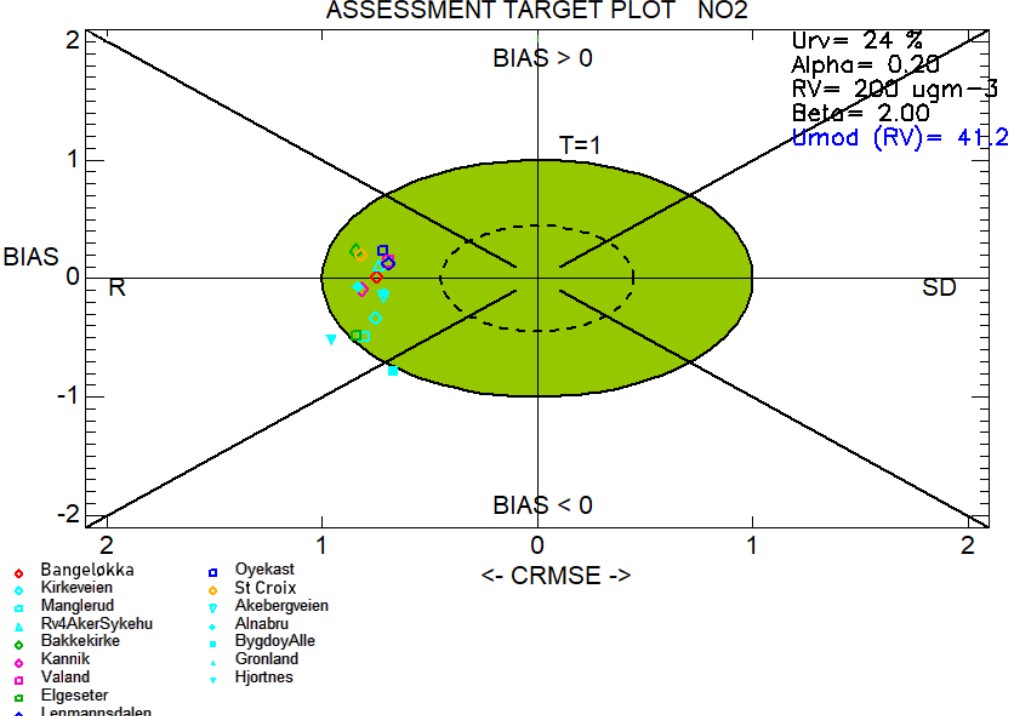

**Figure 11. Target plots created using the winter only (December, January, and February) hourly time series of NO₂ concentrations for all sixteen in-situ stations used for the model evaluation across all six domains. The symbols are colour coded according to each model domain where Drammen is red, Oslo is cyan, Trondheim is green, Stavanger is pink, Grenland is dark blue, and Nedre Glomma is orange.**



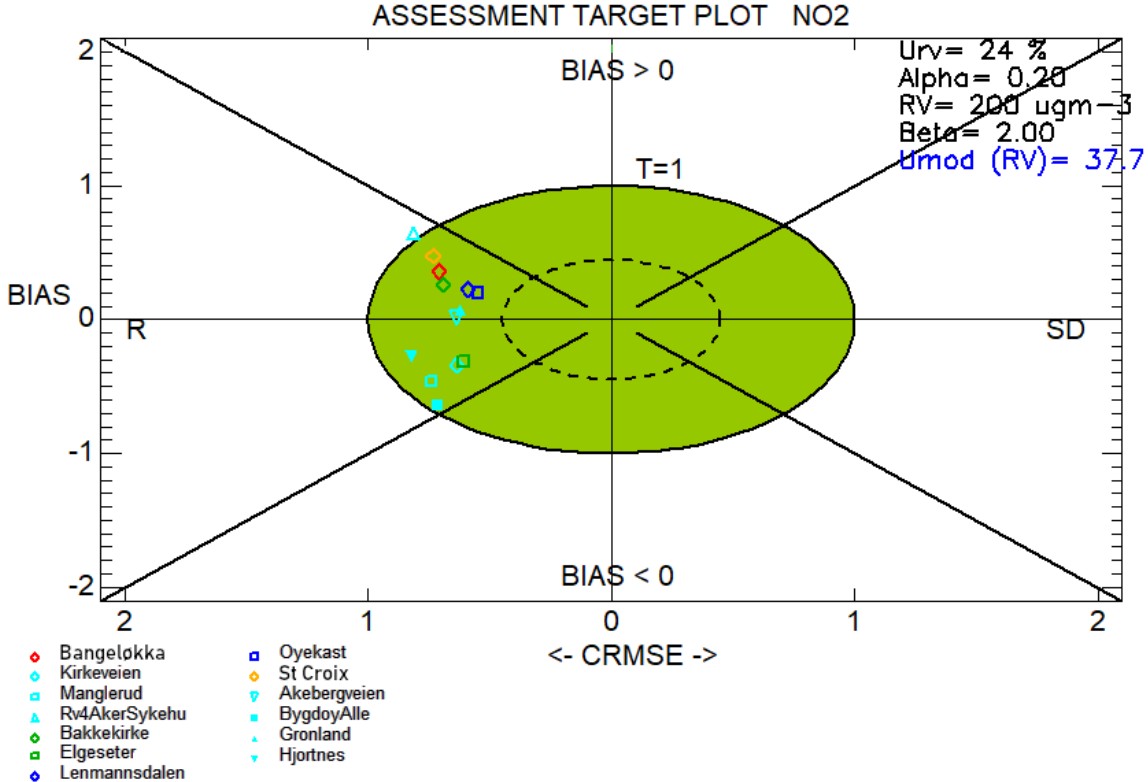

**Figure 12. Target plots created using the summer only (June, July, and August) hourly time series of NO₂ concentrations for thirteen in-situ stations used for the model evaluation across five out of the six domains (excluding Stavanger). The symbols are colour coded according to each model domain where Drammen is red, Oslo is cyan, Trondheim is green, Stavanger is pink, Grenland is dark blue, and Nedre Glomma is orange. Note that there are 3 missing stations (Rv4 AkerSykhus, Kannik, and Våland) during the summer analysis due to insufficient data.**



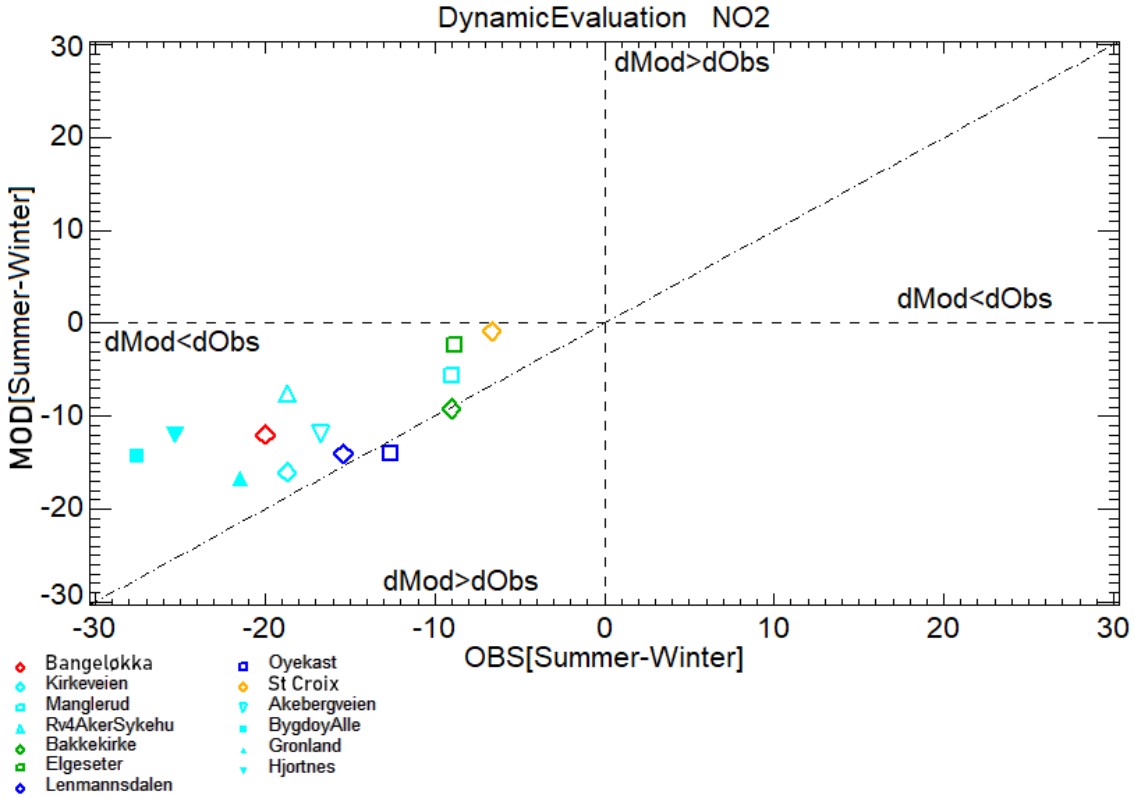

**Figure 13. Evaluation of the mean summer minus winter differences in NO₂ concentration for thirteen out of the sixteen in-situ stations. The missing stations, Kannik, Våland, and Alnabru, have insufficient observations during the summer to perform this evaluation in a robust manner.**





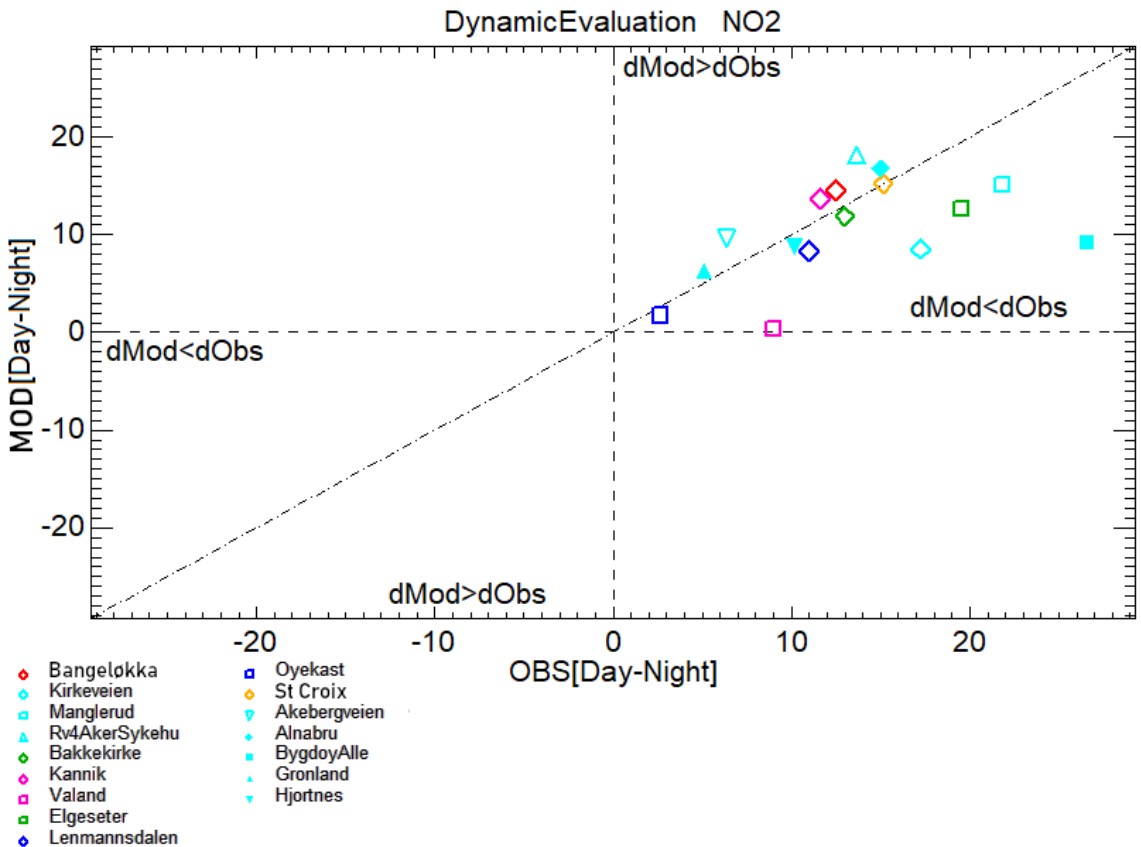

**Figure 14. Evaluation of the mean day minus night differences in NO₂ concentration for all sixteen in-situ observation stations. For the purposes of this evaluation, a constant period defining day (8h-19h) and night (20h-7h) are used throughout the year.**



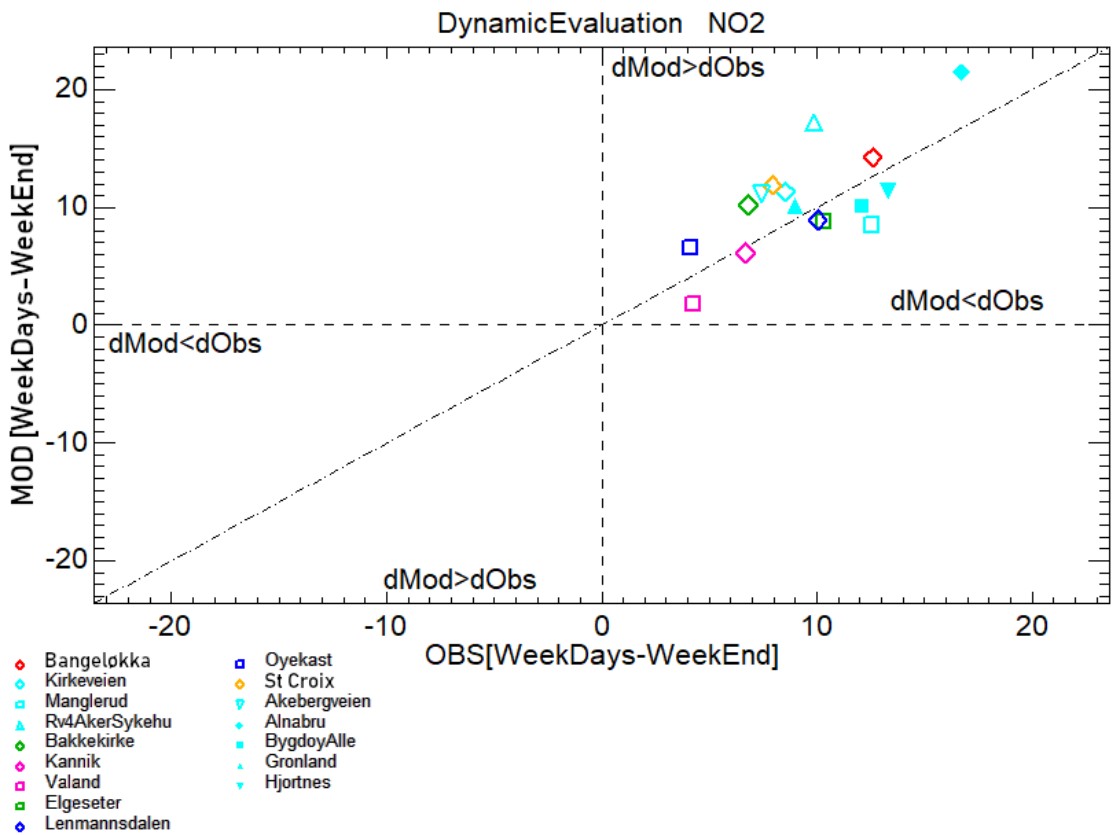

**Figure 15. Evaluation of the mean weekday minus weekend NO₂ concentrations for all sixteen in-situ stations.**



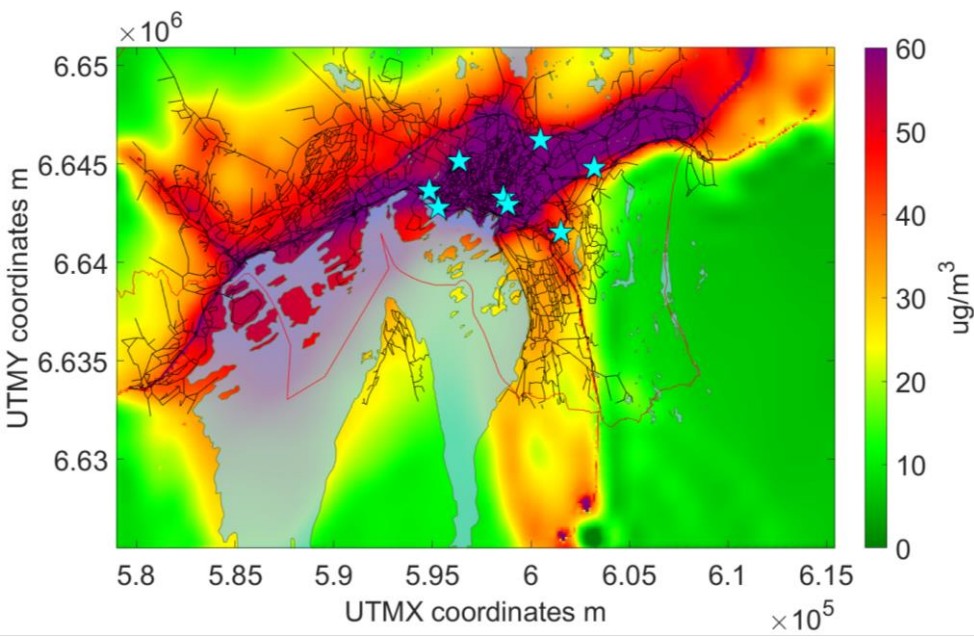

**Figure 16. Simulated daily mean NO₂ concentrations for December 11ᵗʰ from the EPISODE model over the Oslo domain at 100 m x 100 m spatial resolution. This day was selected from a pollution episode lasting from December 9ᵗʰ until December 13th. The concentrations are derived from the receptor point concentrations and then re-gridded onto a 100 m grid. The colour scale shows the range in annual mean NO₂ concentrations between 0 and 60 µg m⁻³. The light blue stars indicate the locations of the observation stations. The blue areas represent the sea, lakes and rivers. The red lines represent administrative boundaries, and the black lines are roads. © OpenStreetMap contributors 2019. Distributed under a Creative Commons BY-SA License.**



**Figure 17. Time series of NO₂ concentrations for the (a) Åkebergveien and (b) Manglerud measuring station in Oslo during a pollution episode lasting from December 10th to 13th 2015. Receptor point concentrations from the model are shown in red, and the observed concentrations are shown in blue.**

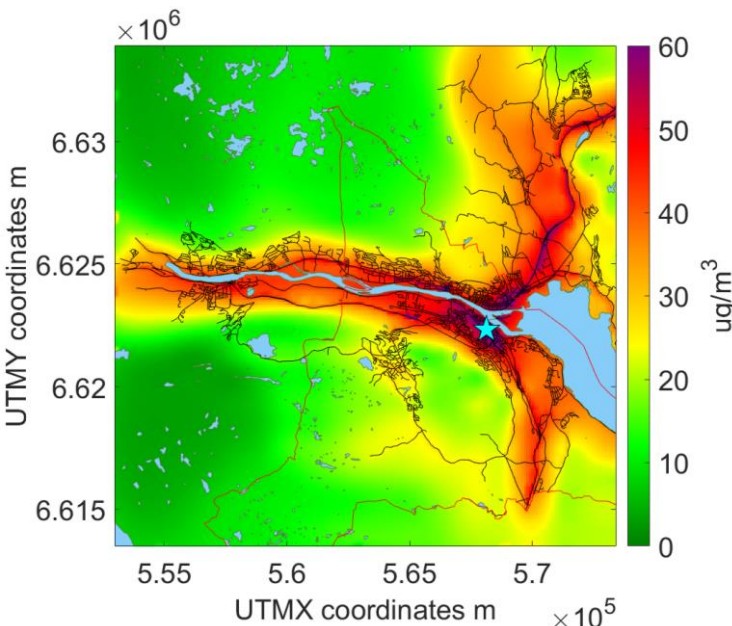

**Figure 18. Simulated daily mean NO₂ concentrations for January 5th from the EPISODE model over the Drammen domain at 100 m x 100 m spatial resolution. This day was selected from a pollution episode lasting from January 4th until January 7th. The concentrations are derived from the receptor point concentrations and then re-gridded onto a 100 m grid. The colour scale shows the range in annual mean NO₂ concentrations between 0 and 60 µg m⁻³. The light blue stars indicate the locations of the observation stations. The blue areas represent the sea, lakes and rivers. The red lines represent administrative boundaries, and the black lines are roads. © OpenStreetMap contributors 2019. Distributed under a Creative Commons BY-SA License.**

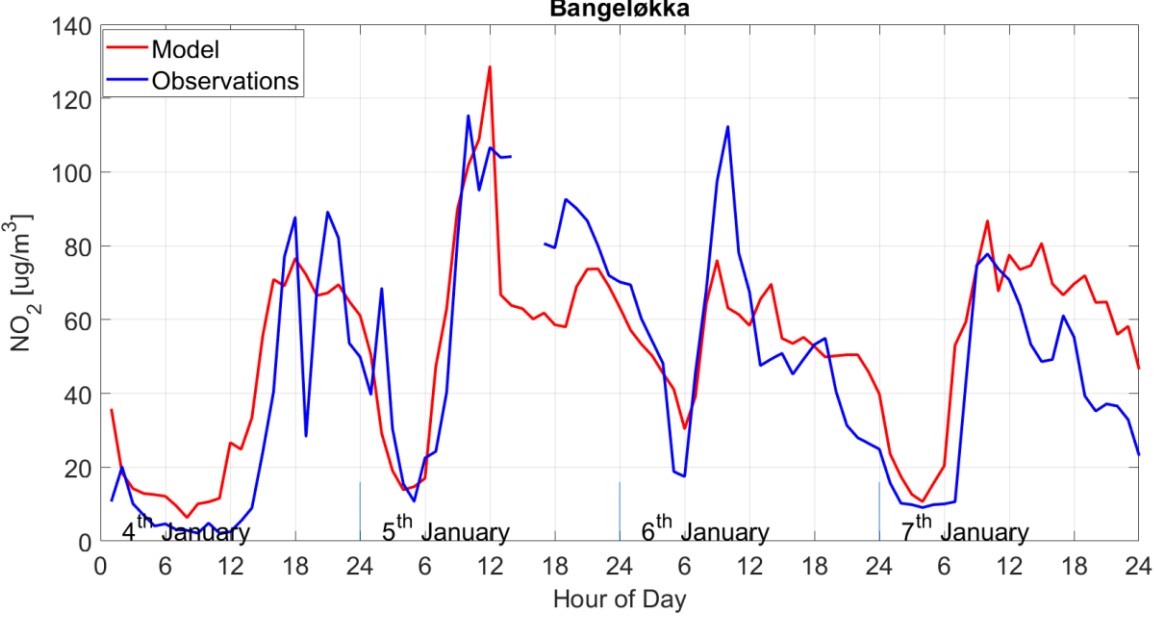

**Figure 19. Time series of NO₂ concentrations for the Bangeløkka measuring station in Drammen during a pollution episode lasting from January 4ᵗʰ to 7ᵗʰ 2015. Receptor point concentrations from the model are shown in red, and the observed concentrations are shown in blue.**

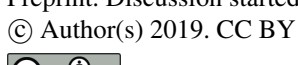



**Table 1. A compilation of all of the possible 3D advection and diffusion schemes usable for the EPISODE Eulerian grid transport.**

| Process | Options | Usage | Description/reference |
|---|---|---|---|
| Horizontal advection | Positive definite 4$^{th}$ degree Bott scheme | Recommended for use in EPISODE | Bott (1989, 1992, 1993) |
| | Positive definite and monotone 4$^{th}$ degree Bott scheme | Experimental, for test purposes only | Bott (1992, 1993) |
| Advection in the vertical | Simple upstream method | Recommended for use in EPISODE | Byun et al. (1999) |
| Horizontal diffusion | Fully explicit forward Euler scheme. | Recommended for use in EPISODE | (Smith, 1985) |
| Vertical diffusion | Semi-implicit Crank-Nicholson diffusion scheme | Recommended for use in EPISODE | Byun et al. (1999) |
| | Urban K(z) method | Newly implemented method, and recommended for specific applications | Beljaars and Holtslag (1991) |

**Table 2. A list and description of all of the possible methods to include initial and background pollutant concentrations in EPISODE model simulations.**

| Method | Temporal Specification | Data Format |
|---|---|---|
| Constant concentration over the entire domain | Constant in time | Set in runfile |
| Constant concentration over the entire domain evolving in time | Hourly | ASCII file |
| Identical concentration column profile covering the entire domain in each vertical layer | Constant or hourly | ASCII file |
| 3D concentration field | Hourly | ASCII file or NetCDF file |





**Table 3. Description of the pre-processing utilities used for preparing input files for the EPISODE model.**

| Pre-processing Utility | Purpose | Required Input | Pre-processing Output |
|---|---|---|---|
| MCWIND | MCWIND creates diagnostically fields of meteorological variables using meteorological observations | Meteorological observations (temperature, wind speed, relative humidity, wind direction, precipitation, and cloud cover) from two or more meteorological observation stations. Requires the observed differential in temperature between two heights in order to infer vertical stability. | Meteorological fields on the EPISODE model horizontal and vertical gridding. All variables can be specified in ASCII or binary format. MCWIND can also create constant topography and surface roughness fields across the entire domain. |
| CAMSBC | Downloads and interpolates the CAMS regional air quality forecasts to the EPISODE modelling domain and grid | Downloaded CAMS regional forecast in NetCDF or GRIB2 formats | Interpolated initial and background concentrations for the EPISODE model domain |
| UECT | UECT produces the various emission input files for point sources, line sources and area source categories independently of AirQUIS | Emission data of geo-referenced or gridded yearly emission totals for $NO_x$, NMVOC, CO, SO2, NH3, PM2.5 and PM10 in a tabular CSV file | Emission input files in ASCII-format for EPISODE containing hourly varying emission data defined for each source category and pollutant |
| TAPM4CC | TAPM4CC creates 2-D and 3-D meteorological fields based on output from the TAPM model | TAPM *.outa file of a simulation with the number of vertical layers matching that of the EPISODE model domain | Hourly meteorological 2-D and 3-D (24 vertical layers up to 3750 m height) and topography input files in binary format for use in EPISODE |
| Auxiliary utilities | Utilities for creating topography and surface roughness input files for EPISODE. | One can either extract the topography and surface roughness from the WRF and AROME meteorological files, or you can specify constant values across the domain | Input files of surface roughness and topography in ASCII format for the EPISODE model domain (only relevant when running with AROME meteorology) |



**Table 4. A description of the data sources, the methodology used, and the reference years for the emission inventories for each emission sector used in the case studies. NRA: Norwegian Road Administration, OFV: Opplysningsrådet for Veitrafikken. HBEFA: Handbook Emission Factors for Road Transport. NCA: Norwegian Coastal Administration. NPRTR: Norwegian Pollutant Release and Transfer Registers.**

| Emission Sector | Data Source | Methodology | Reference Year |
|---|---|---|---|
| On road | NRA (ADT), HBEFA (EF), OFV (Vehicle fleet technology composition) | Traffic emission model | 2013 |
| Off road | Statistics Norway | Statistics at the district level and gridding using GIS software | Drammen (2012), Oslo (1995), Stavanger (1998), Trondheim (2005) |
| Shipping | NCA, except in Oslo, for which it was used data provided by the Port of Oslo and NILU databases described in López-Aparicio et al., 2017 | AIS and Activity data (Oslo) | 2013 |
| Industrial | Statistics Norway, facility level and NPRTR | Emission officially reported by entities or estimated based on data from facilities | Drammen (2012), Grenland (1991/2015), Nedre Glomma (2012), Oslo (2013), Stavanger (1998/2015), Trondheim (2005/2015) |

**Table 5. A description of the emission type and the percentage emission of $NO_x$ as $NO_2$ (as NO2 mass equivalent) for each sector considered in the model simulation case studies.**

| Emission Sector | Emission Type | Percentage emission of $NO_x$ as $NO_2$ in terms of $NO_2$ mass equivalent |
|---|---|---|
| On road | Line source | Varying between 4.5% to 45.9% (with an approximate mean of 15%) |
| Off road | Area source | 10% |
| Shipping | Area source | 10% |
| Industrial | Area source (point sources in Grenland) | 10% |



**Table 6. A description of the horizontal extent, vertical gridding (shown as the height at the top and at the mid-level of each layer) with the mid-level points shown in brackets) and the number of receptor points for each model domain. Note that identical vertical gridding was used for all six cities.**

| Model domain | Horizontal extent (km × km) | Vertical gridding – Layer tops (m) | Vertical gridding – mid-layer heights (m) | Number of receptor points |
|---|---|---|---|---|
| Oslo | 38 × 27 | 24, 48, 72, 98, 125, 153, 184, 218, 254, 294, 338, 386, 436, 493, 552, 621, 692, 771, 858, 950, 1050, 1157, 1275, 1401, 1538, 1686, 1844, 2016, 2195, 2387, 2591, 2805, 3032, 3270, 3518 | 12, 36, 60, 85, 111.5, 139, 168.5, 201, 236, 274, 316, 362, 411, 464.5, 522.5, 586.5, 656.5, 731.5, 814.5, 904, 1000, 1103.5, 1216, 1338, 1469.5, 1612, 1765, 1930, 2105.5, 2291, 2489, 2698, 2918.5, 3151, 3394 | 34040 |
| Trondheim | 14 × 16 | idem | idem | 10293 |
| Stavanger | 14 × 25 | idem | idem | 16496 |
| Drammen | 23 × 22 | idem | idem | 13758 |
| Grenland | 16 × 23 | idem | idem | 13661 |
| Nedre Glomma | 29 × 22 | idem | idem | 28498 |





**Table 7. Observation stations used in the evaluation of the EPISODE model results for the six different city domains. The location of each station is shown in UTM coordinates along with the corresponding UTM grid.**

| City/Domain | Observation Station | UTM Coordinates (X-UTM,Y-UTM) | | Station Type |
|---|---|---|---|---|
| Oslo | Åkebergveien | 598845, | 6642929 | Traffic |
| | Alnabru | 603212, | 6644794 | Traffic |
| | Bygdøy Alle | 594854, | 6643637 | Traffic |
| | Gronland | 598697, | 6642974 | Urban background |
| | Hjortnes | 595188, | 6642860 | Traffic (high volume) |
| | Kirkeveien | 596377, | 6645131 | Traffic (high volume) |
| | Manglerud | 601533, | 6641533 | Traffic (high volume) |
| | Rv4 Aker Sykehus | 600444, | 6646186 | Traffic (high volume) |
| Drammen | Bangeløkka | 568124, | 6622332 | Traffic (low volume) |
| Nedre Glomma | St Croix | 611082, | 6565092 | Traffic (high volume) |
| Grenland | Lensmannsdalen | 193449, | 6570117 | Traffic (high volume) |
| | Øyekast | 193541, | 6566749 | Influence from industry and harbour |
| Stavanger | Kannik | 311922, | 6540558 | Traffic (high volume) |
| | Våland | 311898, | 6540686 | Urban background |
| Trondheim | Bakkekirke | 570411, | 7034630 | Traffic |
| | Elgeseter | 569691, | 7033059 | Traffic (high volume) |

**Table 8. Mean statistics presented in the Taylor for all sixteen observation stations for the full year, the winter, autumn, and summer**
5 **seasons. $\sigma M/\sigma O$ is the ratio of the model and observed standard deviation in $NO_2$ concentrations, R is the Pearson correlation coefficient, RMSE is the Root-mean squared error, and IOA is the index of agreement. These statistical metrics are explained in further detail in Appendix B.**

| Time Period | $\sigma M/\sigma O$ | R | RMSE | IOA |
|---|---|---|---|---|
| Annual | 1.05 | 0.6 | 0.95 | 0.74 |
| Winter | 0.90 | 0.64 | 0.84 | 0.76 |
| Autumn | 1.16 | 0.62 | 0.98 | 0.74 |
| Summer | 1.11 | 0.5 | 1.09 | 0.65 |




**Table 9. Compiled statistics for the comparison between the observed and modelled NO$_2$ concentrations during the December 10th to 13th pollution episode in Oslo. Statistics for each station are shown along with the mean of all of the statistics.**

| Station | σM/σO | R | RMSE | IOA |
|---|---|---|---|---|
| Alnabru | 0.58 | 0.57 | 0.82 | 0.66 |
| Manglerud | 0.79 | 0.61 | 0.81 | 0.75 |
| Rv4 Aker Sykhus | 0.94 | 0.65 | 0.81 | 0.80 |
| Bygdøy Alle | 0.66 | 0.74 | 0.68 | 0.67 |
| Kirkeveien | 0.85 | 0.75 | 0.67 | 0.78 |
| Gronland | 0.81 | 0.77 | 0.64 | 0.83 |
| Åkebergveien | 0.97 | 0.84 | 0.56 | 0.88 |
| Hjortnes | 0.52 | 0.84 | 0.63 | 0.73 |
| Mean | 0.77 | 0.72 | 0.70 | 0.76 |