# Peer review of "The urban dispersion model EPISODE v10.0. Part 1: A Eulerian and sub-grid-scale air quality model and its application in Nordic winter conditions"

_Geoscientific Model Development, 2019_

## Short Comment (SC1) · 7 Aug 2019

Dear authors,

in my role as Executive editor of GMD, I would like to bring to your attention our Editorial version 1.2:

https://www.geosci-model-dev.net/12/2215/2019/

This highlights some requirements of papers published in GMD, which is also available

on the GMD website in the 'Manuscript Types' section:

http://www.geoscientific-model-development.net/submission/manuscript_types.html

In particular, please note that for your paper, the following requirement has not been met in the Discussions paper:

- "The main paper must give the model name and version number (or other unique identifier) in the title."

Please add the version number for EPISODE in the title upon your revised submission to GMD.

Yours,

Astrid Kerkweg

---

## Referee Comment (RC1) · Anonymous Referee #1 · 20 Sep 2019

**Review for paper:**
*"The urban dispersion model EPISODE. Part 1: A Eulerian and sub- grid-scale air quality model and its application in Nordic winter conditions"*
*by*
*P. D. Hamer et al.*

*Submitted to Geoscientific Model Development*

**General Comments:**
This paper presents a Eulerian urban dispersion model, EPISODE, which consists of a 3D CTM including sub-grid dispersion modules. Such a model is essential for estimating the exposure of urban populations to pollution. First, the authors describe each component of the model, and second, they assess the capability of the model to simulate $NO_2$ levels over six cities in Norway. However, some parts of the paper is not very clear or incomplete or, in my opinion, not necessary regarding the aim of the paper. This leads the paper very long with numerous tables and figures.

**Overall recommendation:**
I recommend that the paper should be accepted for publication in Geoscientific Model Developemnt after major and specific revisions listed below.

**Major Revisions:**
First, I suggest an effort to shorten the paper and limit the number of tables and figures for more clarity.
My second main concern is about the meteorological drive of EPISODE. I understand that the 3D part of EPISODE is a CTM and I think it is necessary to explain how the meteorological inputs are provided to EPISODE and at which temporal frequency. I guess that several options are available. However, Only two are briefly described and one in an obscure way (I don't know what is TAPM).
Third, in my opinion, the assumption of PSS could explain some discrepancies between simulated results and observations. However, this is not discussed except in the last part on future work.

**Specific Revisions:**
**Introduction:**
- The discussion on LES modelling is very short and lacks from citations and examples of obtained results with such models.
- Others models using the same concept than EPISODE have been cited but no comparison is done between them and EPISODE. In particular, the originality of EPISODE compared to these previous models should be assessed.

**Part 2:**
- 2.1: I understand that no chemical evolution of PM2.5 and PM10 is implemented in EPIDOSE but I wonder if microphysical processes (coagulation, sedimentation) are taken into account. At which time-step, the meteorological inputs are given to EPISODE?
- 2.2.1:
    - I understand that horizontal and vertical resolutions are flexible depending on the choice of the user. Could you please give the available range of horizontal resolutions and the typical number of vertical levels?

- Page 7, lines 16-18: the information about topography should be moved page 5 in the first paragraph after 2.2.1 when the vertical grid is detailed.
- Page 8: equations are hard to read, the font is too small. In equation (2), I guess it is $K_*(z)$ and not $K(z)$.
- Page 9, lines 7-10: could you please explain that the surface roughness is needed to compute the friction velocity?
- In table 2, it is indicated that constant concentration profiles are given as ASCII files while it is mentioned in the text (page 10, line 3) that they have to be specified in the EPISODE run file.
- Page 10, line 13: what are the NBV and BedreByLuft projects?
- Page 10, lines 17-18: it is indicated how the background concentrations are provided to EPISODE in the example presented in part two of the article but not for the one presented in part one.
- Page 11, line 6: please indicate how $J(NO_2)$ is computed, in particular the actinic flux.
- Concerning the PSS via R4, the authors should specify that it is adequate in polluted Nordic wintertime conditions especially during the day. Indeed during the night, the $N_xO_y$ (including $N_2O_5$, $NO_3$ and $HNO_3$) chemistry should be dominated.
- 2.2.2: Page 11, lines 28-31: I do not understand how the location of the road links is given to the model.
- 2.3: This section should be carefully read, it is difficult to understand, for instance:
  - UECT is described in two separated paragraphs.
  - What is TAPM?
  - I do not understand how it is possible to use 3D meteorological fields from AROME or WRF in EPISODE. I guess it implies a pre-processing of these fields to use then in EPISODE. Could you please clarify this point? Also at which temporal resolutions, meteorological fields have to be provided to EPISODE? See also major comment for this point.

**Part 3**

The information about the temporal frequencies of meteorological outputs given to EPISODE from AROME is missing. Why do you not use point source emissions? Could you please justify? I suggest adding a figure showing the location of the chosen urban areas including each domain of simulations if possible. The information about the vertical grids and the horizontal domain extend should be given at the beginning of the part and not at the end (table 6).

**Part 4:**

- 4.1.1: I'm not sure that this part provides interesting information regarding the aim of the paper. I suggest deleting it to shorten the paper. If the decision is to conserve it, could you please discuss the interest to provide annual mean concentration maps? Maybe this information could be relevant for abatement strategy?
- 4.1.2: The limitation due to the PSS hypothesis should be discuss in regards to the NxOy chemistry occurring during night (see comments on part 2.2.1).
- 4.1.3: I am not convinced of the interest to look at these kinds of differences. In particular, the use of mean values makes it difficult to separate processes that may explain differences between simulations and observations. Moreover, again, the effect of the PSS hypothesis and of the non-linearity of atmospheric chemistry, which is not taken into account, is not discussed.

- 4.2.1: Could you please give some possible reasons for this polluted event?
- 4.2.2: Same comment as 4.2.1

**Parts 5 and 6:** I suggest combining parts 5 and 6 in a part called "conclusion and future work".

---

## Referee Comment (RC2) · Anonymous Referee #2 · 20 Dec 2019

SC on The urban dispersion model EPISODE. Part 1: A Eulerian and subgrid-scale air quality model and its application in Nordic winter conditions.

The authors present work around a new air quality model. Whilst some theoretical background is covered, the paper is a little difficult to read and request a small number of general and minor points are responded to before publication in GMD.

General points:

It is stated that this model follows the same approach as others. Would an intercom-

parison be possible? I appreciate the clear statement on the PSS assumption, but there is some reference to another model which has a gas phase component - is it not possible to compare outputs from the PSS assumption? Im not sure this assumption is worthwhile keeping for a generically useful model. Please clarify.

Minor points:

The section numbering, and reference to different sections in page 4 is confusing. When you refer to part 1, this is labelled as Section 2.

There are a number of formatting errors in the document. Please revisit the text and correct. These include:

Page 4. 'Sect. 6' please be consistent in using 'Section'

Page 4. line 6 '. part two(Karl et al., 2019) of this article describes the EPISODE-CityChem model'. Is this meant to be a new sentence? Also what article? Karl et al 2019? You then say 'Part two describes an application of EPISODE-CityChem for the city of Hamburg'. Part 2 in this paper? I think it is another paper.

4.1.2 Full-Year and Seasonal Model Evaluation - the formatting is tight spacing and bold, please correct

Page 16: 'The data sources, the methodology used, and emission reference years are summarized in .....Table 4 for each emission sector' there is a large gap please correct.
* * *

---

## Referee Comment (RC3) · Anonymous Referee #3 · 29 Jan 2020

General comments:

The manuscript documents the comprehensive and "much applied" model system EPISODE at NILU. As a model development description paper it merits publications and the presentation of results and the statistical validation is interesting and relevant. However some revisions are needed in which the main issues are:

- In general the paper suffers from too much lengthy and unnecessary descriptions, repetitions and, in some sections, too many details. A more concise language and a

better structuring of the paper is needed. Thus the authors should work out a more concise version before publication to increase readability. Some examples on how the paper can be improved are given in the detailed comments below.

- In the manuscript the model is presented as "new", which is somewhat surprising since the EPISODE model is well known for many project applications in the Nordic areas during the last 15-20 years. It should be made clearer what is new in the present version compared to earlier model descriptions (for example Slørdal et al. , 2003). At the same time it is acknowledged that it is important to publish a model description including new revisions.

Detailed comments:

Abstract:

Page 1, 14: It is somewhat surprising that PM2.5 and PM10 is not included in the paper since the health concerns probably are stronger for these two components, and since the model EPISODE also largely has been applied to PM modelling (as documented in reports from NILU etc.)

Page 2, 2-4: The model seems not to be applicable to a range of policy applications in local air quality, but rather to more specific policy applications involving NOx. Please rewrite this.

Page 2, 2: Replace "...assess of trans-boundary..." to ...assess transboundary...

Page 3, 8-15. References to the model EPISODE is missing here. The model has been applied (but may be not documented in refereed journals) for quite some time. For example gives Slørdal et al. 2003 a quite thorough technical description of the model. Please add references.

Page 3, 23-26. It is rather unclear what the authors mean with micro-scale modeling, it is not necessary to run a LES-model in order to model on the micro-scale. Please define micro-scale properly, or remove.

Page 4, 20. Sentence "Episode consist of ..." repetition of what is said in the introduction, please revise and make the paper more concise (see also general comment above).

Page 4, 29. Explain acronyms NWP, AROME, WRF

Page 5, 1-2, 10-11, 19-20. Examples on unnecessary repetition.

Page 5, 7. The sentence "We also .." appears as an unnecessary statement.

Page 6, 20-21. How is convection solved by bulk transport? Please explain or give a reference to how this is parameterized.

Page 6, 26. ". . .. very low artificial numerical diffusion...". How low? For very steep gradients numerical diffusion should be expected from any Eulerian scheme. Please discuss this issue in more detail and explain how it may affect the simulations close to large sources.

Page 7, 14-15. What about the bulk vertical convection, is this also solved by use of the upstream scheme? Please explain.

Page 7, 20-21. Please explain better what is meant with " ...dependence on spatial structure of the flow field ...".

Page 7, 26. Smith, 1985, is not found in reference list?

Page 7, 32. ". . ..K-theory..." should be " . . . Monin-Obukhov similarity theory. . ...

Page 8, 1-4. Is the vertical profile of K prescribed? K(chem-comp) = K(heat) which I would expect to be found from the meteorological data based on what is previously said in the paper? The descriptions and assumptions in this section needs to be made clearer.

Page 8, 26. It is said "The new urban . . ..", please explain better what is new compared to the description in Slørdal et al. (2003). Also since this is a new parameterization,

refererence to a previous validation or a comparison of the new method to local turbulence observations are missing. Please include.

Page 9, section "Area Gridded Emission". This sections has unnecessary many details, for example the units of the emissions, ASCII format etc., details rather to be entered in user manuals or an appendix.

Page 10, 23-25. How large fraction of the emissions are assumed to be NO2? This is not clearly stated. Diesel engines could have as much as 10-20 % direct emissions of NO2, so if all emissions are NO it should be argued why.

Page 15, Section 2.3. There are lots of details in this section that should be put elsewhere or excluded to improve the readability of the text.

Page 16, section 3. The importance of the paper would have been larger if PM2.5 and PM10 had been included in the case studies.

Page 18. Section 4.1.1. These section also have several unnecessary repetitions and statements, partly "essay style". Please make the text more concise. Just as an example, first sentence of line 15 is clearly unnecessary.

Page 20, 29. Units of RMSE?

Page 22. A discussion of the uncertainties in wintertime NOx emissions from cold engines, and the uncertainties this may imply in the model results, are missing.

Page 25, section 4.2.1 and 4.2.2. A quantitative comparison with local meteorological data (both model data used in the EPISODE model and local measurements) must be given and may shed light on what is happening in these two cases. Please include.

Figures and Tables:

Figures 3-6 are hard to read and must be improved. Geographical information must be added and the different concentration classes on the maps must be made clearer. The same applies to Figures 16 and 18, although the concentration levels are more clearly

seen in these figures. Also, for the time-series, avoid legends overlaying the curves.

Apart from this the Figures and Tables are satisfactory.

---

## Author Comment (AC1) · 16 Jun 2020

Thank you very much for pointing out this error. We have now added the version number (v10) into the paper title.
* * *

---

## Author Comment (AC2) · 16 Jun 2020

**Responses to Reviewer #1**

We have written a very long manuscript, so we would first like to thank reviewer #1 for taking the time to review our paper and for giving a lot of very useful criticism on different aspects of the paper. We believe that their comments will lead to a significant improvement in its quality. We have quoted the relevant text from the review (shown in italics) and have responded below each comment in times new roman. In addition, we have prepared a revised manuscript showing the revised text in red.

**Major Revisions**

*First, I suggest an effort to shorten the paper and limit the number of tables and figures for more clarity.*

We thank the reviewer for identifying this concern. We have attempted to address this issue in a number of ways:

- First by filtering the entire document to check for material that can be removed.
- Simplifying the style of the writing as much as possible.
- Section 2.4 (previously 2.3) now relies more on Table 3 to present the information and all repetitions are removed between the text and table and all duplicate information has been removed from the text.
- By following the specific comments of the reviewer about Sect. 4.1.3: we have removed this entire section and all of its associated figures.

*My second main concern is about the meteorological drive of EPISODE. I understand that the 3D part of EPISODE is a CTM and I think it is necessary to explain how the meteorological inputs are provided to EPISODE and at which temporal frequency. I guess that several options are available. However, Only two are briefly described and one in an obscure way (I don't know what is TAPM).*

We thank the reviewer for identifying this problem. Having reviewed the manuscript we completely agree with their comment that the manuscript lacks enough detail on this point. We have therefore modified the manuscript in several locations to correct this problem. The relevant sections in the new document are:

- There is a new section '2.3 Meteorological Inputs' that details the information relevant for the input meteorology.
- Within section 2.3 we have now properly referenced TAPM and also given specific direction to consult Part Two of this paper where TAPM is more thoroughly described and discussed in a specific implementation example.
- Also, in section 2.3, we have now more clearly explained the AROME meteorological fields are available from Met Norway's THREDDS server. We give a clearer link to the supplement that describes these data in detail. In addition, we clearly explain the frequency at which the data are available.
- We also describe the use of the WRF meteorological data.

*Third, in my opinion, the assumption of PSS could explain some discrepancies between simulated results and observations. However, this is not discussed except in the last part on future work.*

We thank the reviewer for identifying this issue. Although we did try to highlight the limitation of the PSS during summer in Sect 4.1.2 (lines 21-27, page 21) in the original manuscript this was perhaps not made clear enough. We now state this more clearly. Indeed, reviewer #2 has made similar criticisms, and we therefore have drafted a common response to both sets of comments. We have modified the manuscript text in several locations to expand the discussion of the PSS and how it affects our results. The new discussion aims to justify the PSS assumption, yet also highlight its limitations both in the

Nordic context and in other locations. These discussions are in Sect. 4.1.2 (lines 26 onwards, page 20) and in the summary.

Specific Revisions

Introduction

*The discussion on LES modelling is very short and lacks from citations and examples of obtained results with such models.*

We have now added appropriate description of microscale modelling methods and have provided literature examples discussing LES methods.

*Others models using the same concept than EPISODE have been cited but no comparison is done between them and EPISODE. In particular, the originality of EPISODE compared to these previous models should be assessed.*

We have now added a sentence explaining that EPISODE was originally developed at a similar point in time (i.e., 1980s) to models following a similar philosophy, e.g., AirGIS, and that therefore at the point of its original inception it was consistent with the state of the art at that time.

Part 2

2.1

*I understand that no chemical evolution of PM2.5 and PM10 is implemented in EPIDOSE but I wonder if microphysical processes (coagulation, sedimentation) are taken into account. At which time-step, the meteorological inputs are given to EPISODE?*

We are currently at the early stages of implementing sedimentation and below cloud wet scavenging into EPISODE. In order to improve the representation of the physical removal processes, we will also implement size bins to capture the different physical processes affecting the washout of different size modes of particles, e.g., impaction, diffusion, and interception. Although coagulation and other types of particle growth are not currently planned in this round of work, these are processes that we would wish to add in the future. These planned/in-progress developments are now described in the section describing future work.

2.2.1

*I understand that horizontal and vertical resolutions are flexible depending on the choice of the user. Could you please give the available range of horizontal resolutions and the typical number of vertical levels?*

We have now included a description of the ranges in horizontal and vertical resolution that we typically use in Sect 2.2.1.

*Page 7, lines 16-18: the information about topography should be moved page 5 in the first paragraph after 2.2.1 when the vertical grid is detailed.*

We have followed this advice and moved the text to the suggested location.

*Page 8: equations are hard to read, the font is too small. In equation (2), I guess it is K\*(z) and not K(z).*

We have increased the font size to 11 pt to make the equations more readable.

*Page 9, lines 7-10: could you please explain that the surface roughness is needed to compute the friction velocity?*

We have now added this as a note in the description of u*.

*In table 2, it is indicated that constant concentration profiles are given as ASCII files while it is mentioned in the text (page 10, line 3) that they have to be specified in the EPISODE run file.*

The constant concentration profiles have to be specified in separate ASCII files that are referenced from the runfile. We have now correctly described the text to reflect this.

*Page 10, line 13: what are the NBV and BedreByLuft projects?*

We have now included a clearer link to the NBV project and have provided a reference to the BedreByLuft project.

*Page 10, lines 17-18: it is indicated how the background concentrations are provided to EPISODE in the example presented in part two of the article but not for the one presented in part one.*

A discussion of the background data was included in Sect. 3, but we now try to make this clearer. We have now included a short description of how we download the data for the background concentrations in Sect 2.4 as well.

*Page 11, line 6: please indicate how J(NO2) is computed, in particular the actinic flux.*

We do not calculate the JNO2 values using actinic flux from a radiative transfer model. Rather, here we use a 2-parameter scheme to calculate the photolysis rate. The two parameter scheme is already described within the supplement S2 in equation S2.2b. The value of theta in S2.2b is calculated using time of day, date in the year. In addition, the meteorological input variable, cloud fraction, is also used to adjust JNO2. We have now added a reference to this equation in the supplement and main text. We also describe the future work we have planned to upgrade this calculation of JNO2.

*Concerning the PSS via R4, the authors should specify that it is adequate in polluted Nordic wintertime conditions especially during the day. Indeed during the night, the NxOy (including N2O5, NO3 and HNO3) chemistry should be dominated.*

We thank the reviewer for identifying this issue. Although we mentioned briefly our intention to consider N2O5 in the future, we did not give this adequate discussion throughout the paper. We have now added in order to clearly state that this is a limitation.

2.2.2

*Page 11, lines 28-31: I do not understand how the location of the road links is given to the model.*

The location of the road links is specified in a separate ASCII file giving the UTM coordinates of the road link beginning and end points, and the width, and height at the beginning and end points. We have now made this clearer in the text.

2.3

*This section should be carefully read, it is difficult to understand, for instance: o UECT is described in two separated paragraphs.*

We have now removed most of the text here since it was duplicated within Table 3 including the text relating to UECT.

*What is TAPM? o I do not understand how it is possible to use 3D meteorological fields from AROME or WRF in EPISODE. I guess it implies a pre-processing of these fields to use then in EPISODE. Could you please clarify this point? Also at which temporal resolutions, meteorological fields have to be provided to EPISODE? See also major comment for this point.*

We have now described TAPM more clearly (in Sect. 2.4) and have directed readers to Part Two of this paper (Karl et al., 2019) where a more thorough description of TAPM and its uses is given. The details regarding the spatiotemporal resolutions of the input meteorology are also explained in the new Sect. 2.3 on meteorological inputs.

Part 3

*The information about the temporal frequencies of meteorological outputs given to EPISODE from AROME is missing.*

We thank the reviewer for identifying this error. We have now added relevant text describing this.

*Why do you not use point source emissions? Could you please justify?*

We did use point source emissions in the case of the Grenland and Nedre Glomma simulations, but this was not properly explained. Both areas have a particular concentration of industry, and the emissions from the point sources happen to be relatively well characterized. We have now altered the text to describe the point sources in Grenland.

In the cases of the other cities, while there are point sources there in reality, in these cases we lacked the detailed information on the point sources (e.g., stack height, gas flue speed and temperature) to be able to represent these sources with this method.

*I suggest adding a figure showing the location of the chosen urban areas including each domain of simulations if possible. The information about the vertical grids and the horizontal domain extend should be given at the beginning of the part and not at the end (table 6).*

We thank the reviewer for this suggestion. We have now created a new figure (Fig. 3) that plots the locations of the different urban areas on a map of the southern half of Norway.

Part 4

4.1.1

*I'm not sure that this part provides interesting information regarding the aim of the paper. I suggest deleting it to shorten the paper. If the decision is to conserve it, could you please discuss the interest to provide annual mean concentration maps? Maybe this information could be relevant for abatement strategy?*

The annual mean $NO_2$ concentration is one of the EU limit values for this pollutant and so we therefore wish to keep these maps as part of the model analysis and evaluation.

4.1.2

*The limitation due to the PSS hypothesis should be discuss in regards to the NxOy chemistry occurring during night (see comments on part 2.2.1).*

We thank the reviewer for raising this point. We have now added relevant discussion here and highlight the limitations.

4.1.3

*I am not convinced of the interest to look at these kinds of differences. In particular, the use of mean values makes it difficult to separate processes that may explain differences between simulations and observations. Moreover, again, the effect of the PSS hypothesis and of the non-linearity of atmospheric chemistry, which is not taken into account, is not discussed.*

We have now removed this section and the relevant figures. Thank you for this recommendation.

4.2.1 and 4.2.2

*Could you please give some possible reasons for this polluted event?*

And… *Same comment as 4.2.1*

We have now included a comparison of the meteorology in the supplement and we use this to provide a more detailed context of the conditions leading to the worsening of pollution during these events, i.e., cold conditions with low wind speeds.

Parts 5 and 6

*I suggest combining parts 5 and 6 in a part called "conclusion and future work".*

We respectfully disagree with the reviewer. We think that this allows a clearer connection between certain arguments made during the paper regarding the limitations of the PSS and chemistry and the future work that we outline.

---

## Author Comment (AC3) · 16 Jun 2020

**Response to Reviewer #2**

We have written a very long manuscript, so we would first like to thank reviewer #2 for taking the time to review our paper and for giving very useful criticism on different aspects of the paper. Their comments have lead to improvements in the manuscript. We have quoted the relevant text from the review (shown in italics) and have responded below each comment in times new roman. In addition, we have prepared a revised manuscript showing the revised text in red.

General Points

*It is stated that this model follows the same approach as others. Would an inter-comparison be possible?*

We do not have access to other models or the necessary expertise to run them for the same specific case studies unfortunately. In addition, this would significantly widen the scope of an already long paper. That said, this would be interesting to look into and with the appearance of standardized evaluation criteria, i.e., the DELTA tool, this becomes easier. We have mentioned this as an item for future work.

*I appreciate the clear statement on the PSS assumption, but there is some reference to another model which has a gas phase component - is it not possible to compare outputs from the PSS assumption? I'm not sure this assumption is worthwhile keeping for a generically useful model. Please clarify.*

We do refer to another model with a more advanced photochemical scheme. This model is the EPISODE-CityChem model, an extension to EPISODE described in the second part of this two-part paper Karl et al., 2019. EPISODE-CityChem includes different possibilities for photochemical mechanisms that include the PSS as well as more comprehensive mechanisms. Karl et al., 2019 describe a comparison made between the PSS and the EmChem09 photochemical mechanism (70 compounds, 67 thermal reactions and 25 photolysis reactions). The results from this comparison show that photochemical ozone production is very small in the vicinity of highly trafficked streets and motorways; suggesting that the PSS assumption is valid close to sources of NOx pollutions. The highest O3 concentration difference between the PSS and EmChem09 occurred in the outflow of polluted air from the city, implying that advanced photochemistry is necessary for the accurate prediction of O3 in the urban background. We have now made reference to these results within the discussion of the results.

We want to argue that the complexity of chemistry should relate to the model's application. The PSS approximation seems appropriate if one is mainly interested in NO2 within polluted the urban areas. We have now made this clearer within the discussion in Sect. 4.1.2 (lines 26 onwards, page 20) and in the summary.

Minor Points

*The section numbering, and reference to different sections in page 4 is confusing. When you refer to part 1, this is labelled as Section 2.*

I think that confusion has been created on this page due to the way we have referred to the companion paper to this article, part two/Karl et al. 2019, and the fact we initially say "This article consists of two parts.". We had meant this to mean a two-part paper. We have really tried to make this much clearer and have removed all phrases that could be misinterpreted.

*There are a number of formatting errors in the document. Please revisit the text and correct. These include: Page 4. 'Sect. 6' please be consistent in using 'Section'*

The formatting requirements from GMD/Copernicus require that the abbreviations "Fig." and "Sect." be used in running text, and "Figure" and "Section" be used at the beginning of a sentence. We are

constrained by the formatting requirements here. However, we have identified some other technical errors after carefully re-reading the manuscript, so thank you for this recommendation.

*Page 4. line 6 '. part two (Karl et al., 2019) of this article describes the EPISODECityChem model'. Is this meant to be a new sentence?*

Yes, it was a new sentence. Thank you, we have fixed this problem now.

*Also what article? Karl et al 2019? You then say 'Part two describes an application of EPISODE-CityChem for the city of Hamburg'. Part 2 in this paper? I think it is another paper.*

"Part two" refers to the second part of this two-part paper, which is Karl et al., 2019. As we mentioned above, we have described this in an unclear manner in such a way that left it open to interpretation. We have therefore rewritten these descriptions to be more clear.

*4.1.2 Full-Year and Seasonal Model Evaluation - the formatting is tight spacing and bold, please correct*

Thank you, we have now corrected this.

*Page 16: 'The data sources, the methodology used, and emission reference years are summarized in …..Table 4 for each emission sector' there is a large gap please correct.*

Thank you again, we have also corrected this error.

---

## Author Comment (AC4) · 16 Jun 2020

**Response to Reviewer #3**

We have written a very long manuscript, so we would first like to thank reviewer #3 for taking the time to review our paper and for giving a lot of very useful criticism on different aspects of the paper. Their lead to a significant improvement in its quality. We have quoted the relevant text from the review (shown in italics) and have responded below each comment in times new roman. In addition, we have prepared a revised manuscript showing the revised text in red.

General comments

*In general the paper suffers from too much lengthy and unnecessary descriptions, repetitions and, in some sections, too many details. A more concise language and a better structuring of the paper is needed. Thus the authors should work out a more concise version before publication to increase readability. Some examples on how the paper can be improved are given in the detailed comments below.*

We thank the reviewer for this advice. We have revised the paper to remove the unnecessary text and repetitions. We have tried to make sentences as concise as possible. We have also removed many of the unnecessary details within the paper. Lastly, we followed advice from another reviewer and have removed the section analyzing daytime/nighttime, weekend/weekday, and summer/winter differences.

*In the manuscript the model is presented as "new", which is somewhat surprising since the EPISODE model is well known for many project applications in the Nordic areas during the last 15-20 years. It should be made clearer what is new in the present version compared to earlier model descriptions (for example Slørdal et al., 2003). At the same time, it is acknowledged that it is important to publish a model description including new revisions.*

We have now made it clearer in the introduction that a primary motivation to publish this article is to provide a comprehensive and definitive peer-reviewed description of the current version of the EPISODE model, i.e., version 10.0. Further, we make clear in section 2 that version 10.0 bases much of its heritage on the EPISODE version described in Slørdal et al. 2003 and have documented the key advancements in v10.0.

Detailed comments

Abstract

*Page 1, 14: It is somewhat surprising that PM2.5 and PM10 is not included in the paper since the health concerns probably are stronger for these two components, and since the model EPISODE also largely has been applied to PM modelling (as documented in reports from NILU etc.)*

We acknowledge that PM is a very important pollutant due to its significant health impacts. There are a few reasons PM was excluded from the case study in Sections 3 and 4 of the paper:

- Work prior to the submission of the manuscript identified problems with missing pollution source processes, i.e., road dust resuspension and domestic heating emissions linked to meteorology. The addition of both emission processes was planned to be documented in separate more focused research papers.
- We have several planned upgrades to the model representation of PM that we believe will significantly improve the simulations. These include PM below cloud scavenging, sedimentation, and the inclusion of different PM size bins.

We plan to carry out a case study focused on PM in the near future after we have completed the proposed upgrades and this will involve the new emission processes for road dust resuspension and domestic heating.

Despite this, we should make it clearer that the capability to simulate PM is included in the current implementation of EPISODE, and that all of the model components, barring the PSS, are relevant for simulation of PM as well.

*Page 2, 2-4: The model seems not to be applicable to a range of policy applications in local air quality, but rather to more specific policy applications involving NOx. Please rewrite this.*

Thank you for this recommendation. We have altered the text to express this more specific application.

Main body

*Page 2, 2: Replace "…assess of trans-boundary…" to …assess transboundary…*

We have correct this error, thank you.

*Page 3, 8-15. References to the model EPISODE is missing here. The model has been applied (but may be not documented in refereed journals) for quite some time. For example gives Slørdal et al. 2003 a quite thorough technical description of the model. Please add references.*

We have now added this reference. Previously it was cited later in the paper but not included in the reference list.

*Page 3, 23-26. It is rather unclear what the authors mean with micro-scale modeling, it is not necessary to run a LES-model in order to model on the micro-scale. Please define micro-scale properly, or remove.*

We have modified the text here in order to explain micro-scale and give examples of these methods.

*Page 4, 20. Sentence "Episode consist of …" repetition of what is said in the introduction, please revise and make the paper more concise (see also general comment above).*

We thank the reviewer for this recommendation and have acted upon it along with other changes to reduce redundancies. Specifically, we have moved details presented in the introduction into section 2.

*Page 4, 29. Explain acronyms NWP, AROME, WRF*

We have now explained these acronyms. Please note that NWP was defined already on page 3.

*Page 5, 1-2, 10-11, 19-20. Examples on unnecessary repetition. Page 5, 7. The sentence "We also .." appears as an unnecessary statement.*

We have now addressed these examples and also carried out a more extensive revision of the document to remove redundancies.

*Page 6, 20-21. How is convection solved by bulk transport? Please explain or give a reference to how this is parameterized.*

We provide an example for the case using AROME meteorology. At the 1 x 1 km scale of the AROME meteorological simulations it is possible to resolve individual deep convective on the bulk Eulerian grid. We have now included a reference to support this. Shallow convection is represented within

AROME using a parameterization, and we have now included a reference to this scheme. Thus, the wind fields provided by AROME already include vertical motions due to convection.

*Page 6, 26. ": : :. very low artificial numerical diffusion...". How low? For very steep gradients numerical diffusion should be expected from any Eulerian scheme. Please discuss this issue in more detail and explain how it may affect the simulations close to large sources.*

Consistent with the original Bott 1989 paper, we have now changed "very low" to small and we have now given a quantitative explanation of small, i.e., <1% with reference to Bott 1989. We have noted that numerical diffusion will occur with very steep concentration gradients, e.g., close to large sources.

*Page 7, 14-15. What about the bulk vertical convection, is this also solved by use of the upstream scheme? Please explain.*

It would be more precise to say that the upstream scheme does not solve convection but solves the vertical motions of tracers based on the three-dimensional wind fields, which includes both shallow and deep convection. As noted earlier, both convection processes are calculated within AROME and the resulting meteorological wind fields therefore include these motions. The upstream method implicitly assumes that there is no net divergence or convergence within the three-dimensional field, and it is therefore used to ensure full consistency and mass conservation during an EPISODE simulation. However, the parameterizations and treatments of advection within AROME should produce wind fields with no net convergence or divergence and that conserve mass and momentum.

*Page 7, 20-21. Please explain better what is meant with " ...dependence on spatial structure of the flow field ...".*

To simplify this and make it clearer we now state "depends on the properties of the flow field".

*Page 7, 26. Smith, 1985, is not found in reference list?*

We have now added this to the reference list.

*Page 7, 32. ": : :.K-theory..." should be " : : : Monin-Obukhov similarity theory: : :..*

We thank the reviewer for this recommendation and have changed the text accordingly.

*Page 8, 1-4. Is the vertical profile of K prescribed? K(chem-comp) = K(heat) which I would expect to be found from the meteorological data based on what is previously said in the paper? The descriptions and assumptions in this section needs to be made clearer.*

The original wording in the manuscript was not very precise. Prescribed is a poor choice of wording, and we now make it clear that Kz is reconstructed indirectly from the input meteorology via estimation of the Monin-Obukhov length, L and the friction velocity, u*. We do indeed assume that K(chem-comp) is equal to K(heat).

*Page 8, 26. It is said "The new urban : : :..", please explain better what is new compared to the description in Slørdal et al. (2003). Also since this is a new parameterization, refererence to a previous validation or a comparison of the new method to local turbulence observations are missing. Please include.*

We are planning to carry out a comprehensive and focused evaluation of the new urban Kz in a dedicated separate study in the near future. This is dependent on obtaining suitable observations, which we plan on gathering at the earliest opportunity. We have now explained this planned future work in section 6.

We regret that it is not possible to provide an evaluation of each feature of EPISODE presented here but we have had to make compromises in the choices of what to present.

*Page 9, section "Area Gridded Emission". This sections has unnecessary many details, for example the units of the emissions, ASCII format etc., details rather to be entered in user manuals or an appendix.*

While we agree with the overall aim of reducing the length of the paper and improving its readability, one of the reviewers has asked for more detailed information in the section on line sources. We have therefore made a compromise and moved all of the details relevant to the emission input files to a new appendix rather than leaving this information to a manual.

*Page 10, 23-25. How large fraction of the emissions are assumed to be NO2? This is not clearly stated. Diesel engines could have as much as 10-20 % direct emissions of NO2, so if all emissions are NO it should be argued why.*

This information is already expressed in Table 5, but we have now tried to make this clearer in the text at the point referred to.

*Page 15, Section 2.3. There are lots of details in this section that should be put elsewhere or excluded to improve the readability of the text.*

We thank the reviewer for this recommendation and have now removed many of the details here to improve the readability.

*Page 16, section 3. The importance of the paper would have been larger if PM2.5 and PM10 had been included in the case studies.*

We agree. However, as we explain the future work section, at the time of this work there were too many strong limitations on the processes (both emission and loss) governing PM. We prefer to address these concerns in future work.

*Page 18. Section 4.1.1. These section also have several unnecessary repetitions and statements, partly "essay style". Please make the text more concise. Just as an example, first sentence of line 15 is clearly unnecessary.*

We thank the reviewer for this recommendation and have now improved the text to improve the readability.

*Page 20, 29. Units of RMSE?*

We have now corrected this.

*Page 22. A discussion of the uncertainties in wintertime NOx emissions from cold engines, and the uncertainties this may imply in the model results, are missing.*

The NOx traffic emissions do not consider cold start discussion, so we have now added a short discussion on this specifically with regard to the low biases identified in Oslo.

*Page 25, section 4.2.1 and 4.2.2. A quantitative comparison with local meteorological data (both model data used in the EPISODE model and local measurements) must be given and may shed light on what is happening in these two cases. Please include.*

We have now added a quantitative comparison of the local meteorology at Drammen (Berskog) and Oslo (Blindern) on both events to provide additional context within a new section in the supplement (S8). This evaluation includes comparisons between the observed and simulated wind and temperature data at the two selected observation stations. We have also added a short discussion on these comparisons within 4.2.1. and 4.2.2.

Figures and Tables

*Figures 3-6 are hard to read and must be improved. Geographical information must be added and the different concentration classes on the maps must be made clearer. The same applies to Figures 16 and 18, although the concentration levels are more clearly seen in these figures. Also, for the time-series, avoid legends overlaying the curves. Apart from this the Figures and Tables are satisfactory.*

We have now remade the mapping figures to improve the colour scale and the geographical information presented (labels are now included for roads and important geographical features). The colour scale now shows the different concentration classes more clearly. In addition, we have now highlighted exceedances above the 40 ug m-3 annual mean limit value.